# On the role of non-linear latent features in bipartite generative neural networks

Tony Bonnaire
*Laboratoire de Physique de l'École normale supérieure,
ENS, Université PSL, CNRS, Sorbonne Université,
Université Paris Cité, F-75005 Paris, France.*

Giovanni Catania[*]
*Departamento de Física Teórica, Universidad Complutense de Madrid, 28040 Madrid, Spain.*

Aurélien Decelle[†]
*Departamento de Física Teórica, Universidad Complutense de Madrid, 28040 Madrid, Spain. and
Escuela Técnica Superior de Ingenieros Industriales, Universidad Politécnica de Madrid,
Calle de José Gutiérrez Abascal 2, Madrid 28006, Spain.*

Beatriz Seoane
*Departamento de Física Teórica, Universidad Complutense de Madrid, 28040 Madrid, Spain.*
(Dated: August 21, 2025)

We investigate the phase diagram and memory retrieval capabilities of bipartite energy-based neural networks, namely Restricted Boltzmann Machines (RBMs), as a function of the prior distribution imposed on their hidden units—including binary, multi-state, and ReLU-like activations. Drawing connections to the Hopfield model and employing analytical tools from statistical physics of disordered systems, we explore how the architectural choices and activation functions shape the thermodynamic properties of these models. Our analysis reveals that standard RBMs with binary hidden nodes and extensive connectivity suffer from reduced critical capacity, limiting their effectiveness as associative memories. To address this, we examine several modifications, such as introducing local biases and adopting richer hidden unit priors. These adjustments restore ordered retrieval phases and markedly improve recall performance, even at finite temperatures. Our theoretical findings, supported by finite-size Monte Carlo simulations, highlight the importance of hidden unit design in enhancing the expressive power of RBMs.

## I. INTRODUCTION

Statistical modeling plays a central role in modern computational science, especially with the rapid progress of machine learning. Its impact is clearly seen in the success of generative models, which are particularly good at creating and analyzing complex data [1, 2]. Unlike traditional statistical physics—where models are built from known interactions through a Hamiltonian—statistical modeling often works in reverse: it starts from data and aims to infer the underlying parameters that explain it. This inverse approach has proven to be a powerful and flexible way to uncover structure in data and build meaningful models across many areas of science.

Over the past few decades, statistical physicists have made major strides in interpretable modeling, notably by adopting Hinton's Boltzmann Machine [3]—particularly in its version without hidden nodes—to address a wide range of scientific problems [4–7]. Building on a long tradition stemming from the Ising model, researchers have developed a rich set of analytical and numerical tools to efficiently learn interaction parameters, especially pairwise couplings between spins. This work represents a fruitful integration of theoretical understanding and computational techniques [8–15].

However, models like the Ising model, which capture only pairwise interactions and local fields, are inherently limited in their generative power. They can match empirical means and covariances but fail to reproduce higher-order correlations present in real-world data [16–18]. This restricts their ability to model the full complexity of structured datasets. Recent research on bipartite architectures such as Restricted Boltzmann Machines (RBMs)[19] has shown their ability to naturally describe all-order moments based on model parameters. RBMs are structured with two layers: one represents the dataset

---

[*] gcatania@ucm.es
[†] aurelien.decelle@upm.es

variables, and the other contains latent variables for modeling effective interactions. When the hidden units follow a Gaussian distribution, the model effectively reduces to a pairwise interacting system [20, 21]. However, adopting more complex distributions for the hidden units, such as binary or ReLU activations, transforms the RBM into a fully multibody interacting model. This understanding has spurred the development of mappings between RBMs and models of interacting variables that include interaction terms up to potentially any order of interaction [22–26]. Furthermore, these mappings have proven successful in practical modeling and inference applications, evidenced by their effectiveness in recovering models in inverse experiments with both binary [23–25] and Potts variables [26, 27] in visible nodes. This means that by varying the chosen prior distribution for the hidden variables, the RBM transitions from a system dominated by pairwise interactions, similar to the Hopfield/Ising model, to a more complex and powerful formulation that encompasses high order interactions. This combination of expressivity and simplicity in architecture has made RBMs a very appealing tool for data-driven studies in biology [28].

Beyond their practical applications, RBMs have remained a central focus of research in both the physics [29] and computer science [30] communities. Introduced in the early 2000s as tractable generative models [31, 32], RBMs have undergone extensive development. Key advancements include improved training algorithms, such as persistent contrastive divergence [33] and optimized sampling strategies [34–36] for efficient gradient estimation, including novel out-of-equilibrium schemes that enhance generative quality [37, 38]. Progress has also been made on the long-standing challenge of estimating the partition function during learning [36, 39].

From a physics perspective, the learning dynamics and pattern formation in RBMs have revealed rich phenomena, including cascades of phase transitions that give rise to hierarchically structured clusters [40–42]. Their equilibrium properties have also been studied in the framework of associative memory, where the emergence of metastable states has been analyzed using Hebbian learning rules [43, 44]. Recent investigations of RBMs in high-load regimes—where the number of patterns scales with system size—have mapped out their phase diagrams under various hidden-layer priors [45–47]. These studies show that binary hidden units severely limit recall capacity, whereas truncated Gaussian units enhance critical capacity at zero temperature relative to standard Gaussian units [48, 49]. In contrast, in low-load regimes, where the number of patterns is small, successful recall remains possible even with binary hidden units, provided the hidden layer is sufficiently large [50]. Given the importance of the choice of prior, this article proposes to explore how the phase diagram of the model varies in terms of associative memory when changing the prior and depending on the number of latent features.

## II. DEFINITION OF THE MODEL

In this work, we focus on a specific class of bipartite generative neural networks: Restricted Boltzmann Machines (RBMs). These models capture pairwise interactions between two layers of variables: a *visible* layer $\boldsymbol{s} = \{s_i\}_{i=1,\dots,N_{\mathrm{v}}}$, representing the input or dataset, and a *hidden* layer $\boldsymbol{\tau} = \{\tau_\mu\}_{\mu=1,\dots,N_{\mathrm{h}}}$, which encodes potential dependencies among the visible units. Throughout the manuscript, we assume that each visible unit corresponds to a binary variable (i.e., an Ising spin), such that $s_i \in \{-1, 1\}$. The energy function, or Hamiltonian, of the system takes the generic form:

$$\mathcal{H}\left(\boldsymbol{s}, \boldsymbol{\tau}\right) = -\sum_{i\mu} s_i w_{i\mu} \tau_\mu - \sum_{i=1}^{N_{\mathrm{v}}} a_i s_i - \sum_{\mu=1}^{N_{\mathrm{h}}} \mathcal{U}_\mu^h\left(\tau_\mu\right), \quad \text{and} \quad p(\boldsymbol{s}, \boldsymbol{\tau}) = \frac{1}{Z} \exp\left[-\beta \mathcal{H}\left(\boldsymbol{s}, \boldsymbol{\tau}\right)\right], \quad (1)$$

where $\boldsymbol{w} = \{w_{i\mu}\}$ denotes the *weight matrix* connecting visible and hidden nodes in the bipartite graph. The vector $\boldsymbol{a}$ represents local *visible biases* (or magnetic fields, in physical terms), while $\mathcal{U}^h$ is the *hidden potential*, which accounts for prior choices or additional biases on the hidden units. The joint probability distribution $p(\boldsymbol{s}, \boldsymbol{\tau})$ is given by the Boltzmann distribution associated with the Hamiltonian at a given inverse temperature $\beta = 1/T$. Recent studies have shown that the Hopfield model can be mapped onto a bipartite Hamiltonian of the RBM type when the hidden variables are assumed to follow a Gaussian distribution [29, 51, 52]. Specifically, starting from the Hamiltonian of

the Hopfield model and applying the Hubbard–Stratonovich transformation, one obtains:

$$\mathcal{H}_{\text{Hopfield}}\left(\boldsymbol{s}\right) = -\frac{1}{N_{\text{v}}}\sum_{i<j}\sum_{\mu=1}^{P}\xi_i^\mu\xi_j^\mu s_i s_j = -\frac{1}{2N_{\text{v}}}\sum_{\mu=1}^{P}\left(\sum_i\xi_i^\mu s_i\right)^2 + \text{const.}, \tag{2}$$

$$p(\boldsymbol{s}) = \frac{1}{Z}\exp\left(-\beta\mathcal{H}_{\text{Hopfield}}\left(\boldsymbol{s}\right)\right) = \frac{1}{Z}\int\prod_\mu d\tau_\mu \exp\left(-\frac{N_{\text{v}}\beta}{2}\sum_\mu\tau_\mu^2 + \beta\sum_\mu\tau_\mu\sum_i\xi_i^\mu s_i\right), \tag{3}$$

where $P$ represents the number of *patterns*. The exponent appearing inside the integral of (3) corresponds to Eq. (1) with the Gaussian prior $\mathcal{U}_\mu^h\left(\tau_\mu\right) = -\beta N_{\text{v}}\tau_\mu^2/2$. Thus, the Boltzmann distribution can be re-expressed as that of a bipartite model involving the spin variables $\boldsymbol{s}$ and a new set of Gaussian-distributed hidden variables $\boldsymbol{\tau}$. In this reformulation, the weight matrix of the bipartite system corresponds to the set of Hopfield patterns $\{\xi_i^\mu\}$, and, by construction, there are no intra-layer couplings. The equilibrium properties of this model, that we review below, are well understood in the case where the weights $\xi_i^\mu$ are independent, identically distributed binary variables, typically taking values $\pm 1$ with equal probability.

Let us briefly recall the phase diagram of this system. When the number of stored patterns is non-extensive, i.e., $P \sim \mathcal{O}(1)$ with respect to $N_{\text{v}}$, the model undergoes a second-order phase transition at $\beta_{\text{c}} = 1$, similar to the Curie–Weiss model. Below this critical temperature, the system can spontaneously polarize toward one of the stored patterns. Although mixed states (having a non-zero overlap with more than one pattern) can appear when $P$ is odd, they are typically subdominant: their free energy is always higher than that of pure states, which corresponds to the recall of a single pattern.

In the case where the number of patterns is extensive, $P = \alpha N_{\text{v}}$, the phase diagram in the $\alpha - T$ plan is much richer, as can be seen in Figure 1. At high temperatures, the system is in a paramagnetic phase, showing no alignment with the stored patterns—this is indicated by the light-blue region in Figure 1. Upon lowering the temperature, the system undergoes a second-order phase transition into a spin-glass (SG) phase, represented in purple. Within this SG phase, at sufficiently low temperatures and for storage loads $\alpha = P/N_{\text{v}} < \alpha_{\text{c}} \approx 0.14$, a metastable phase emerges (labeled (m)R and shown in yellow), characterized by a strong overlap with one of the patterns. This metastable state becomes thermodynamically dominant through a first-order transition, marked by the boundary between the yellow and red regions. At even lower temperatures and smaller $\alpha$, additional metastable states emerge, exhibiting partial recall of multiple patterns. These are represented by various shades of green in the lower-left corner of the phase diagram.

This model serves as a canonical example of *associative memory*, where the energy landscape becomes shaped by the stored patterns, allowing for retrieval capabilities under appropriate choices of temperature $T$ and load $\alpha$. The *critical capacity*, denoted $\alpha_{\text{c}}$, defines the maximum load at which pattern recall remains possible at zero temperature. In the standard Hopfield model, this value is approximately $\alpha_{\text{c}} \approx 0.14$. The full phase diagram in the $\alpha - T$ plane, including regions of local stability for the recall of $L > 1$ patterns, is shown in Figure 1. Various recent studies have explored strategies to enhance $\alpha_{\text{c}}$ by modifying how patterns are encoded [53–55]. Beyond exact recall, recent work has shed light on how correlations between stored patterns can give rise to generalization [56]. The bipartite formulation of the Hopfield model establishes a direct link with the "classical" Boltzmann Machines by Hinton and collaborators in the 1980s [3, 19]. The Hamiltonian in Eq. 1 (up to an additional magnetic field) is structurally identical to that of the Hopfield model in its bipartite form, with the key difference that the hidden variables are discrete, $\tau_\mu = \pm 1$ [57].

Over the past decade, the phase diagram of RBMs has been extensively investigated [46, 48, 50]. However, much of this work has centered on simplified associative memory models or variants with continuous hidden units—such as Gaussian or ReLU activations—often overlooking the fully binary hidden nodes commonly used in practice. Yet, this discrete formulation, originally proposed in the foundational RBM and still prevalent in modern implementations, remains crucial for both theoretical insights and practical utility.

In the following manuscript, we therefore investigate the behavior of the RBM under various settings. First, using a simple example we illustrate in Section III that the RBM with hidden binary units cannot produce a recall phase because of the weakness of the hidden nodes answer on the visible nodes. We also show how it is possible to recover the typical behavior of the Hopfield model (in the low-load regime) by encoding a pattern on an extensive number of hidden nodes: we further confirm that this is a plausible scenario for the behavior of RBMs when performing a realistic training. In Section IV,

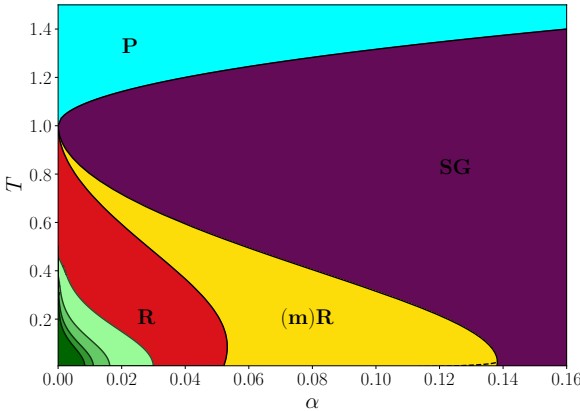

FIG. 1. Phase diagram of the Hopfield model as a function of the temperature $T$ and the density of patterns $\alpha$. P stands for *Paramagnetic*, SG for *Spin Glass*, R for *Recall*, (m)R for *Metastable-Recall*. On top of the recall phase for pure states, we show in shades of green the regions where mixed states with $L$ patterns (for odd $L$ in the range $L \in \{3, 5, 7, 9\}$ from light to dark green) exists as stable local minima of the free energy.

we generalize our analysis to the high-load regime, yet observing very bad recall properties. We then suggest how to improve this behavior by introducing the possibility to silence the hidden nodes in Section V, justifying the relevance of such a procedure also in the case of ReLU-activated hidden nodes.

## III. LOW-LOAD AND BINARY HIDDEN NODES

We begin by considering the case in which both visible and hidden nodes are binary, focusing on the *low-load regime*, where the number of stored patterns is small compared to the system size (i.e., $P \sim \mathcal{O}(1)$). In the Binary-Gaussian RBM (or equivalently, the Hopfield model with a bipartite formulation) standard mean-field techniques from statistical physics can be applied by introducing appropriate order parameters. However, the binary nature of the hidden nodes introduces technical challenges that complicate the analysis. To illustrate the framework, let us define the Hamiltonian corresponding to a *single pattern* $\boldsymbol{w}$ (i.e. $P = 1$) coupled to one binary hidden unit:

$$\mathcal{H}\left(\boldsymbol{s}, \tau\right) = -\tau \sum_i s_i w_i, \tag{4}$$

with $s_i = \pm 1$ and $\tau = \pm 1$. We can express the partition function associated with Eq. (4) by introducing the overlap order parameter $m = N_{\mathrm{v}}^{-1} \sum_i w_i s_i$ (also called *Mattis magnetization*) using a Dirac delta function. After standard manipulations, this yields:

$$
\begin{aligned}
Z &= \sum_{\{\boldsymbol{s}\}} \int dm d\bar{m} \exp\left[\log 2 \cosh\left(\beta N_{\mathrm{v}} m\right) - \mathrm{i} N_{\mathrm{v}} m \bar{m} + \mathrm{i}\bar{m} \sum_i w_i s_i\right] \\
&= \int dm d\bar{m} \exp\left[\log 2 \cosh\left(\beta N_{\mathrm{v}} m\right) - \mathrm{i} N_{\mathrm{v}} m \bar{m} + \sum_i \log 2 \cosh\left(\mathrm{i} w_i \bar{m}\right)\right].
\end{aligned} \tag{5}
$$

where we used that $\sum_{s=-1}^{1} e^{aws} = \sum_{s'=-1}^{1} e^{as'}$ when $w = \pm 1$ with equal probability. The corresponding saddle-point equation for $m$, which determines its value in the thermodynamic limit, is given by

$$m = \tanh\left[\beta \tanh\left(\beta N_{\mathrm{v}} m\right)\right] \underset{N_{\mathrm{v}} \to \infty}{\longrightarrow} \tanh\left[\beta \mathrm{sign}(m)\right] = \mathrm{sgn}(m) \tanh(\beta). \tag{6}$$

The equilibrium behavior of the system determined by the solution of (6) differs significantly from that of the Hopfield model. Specifically, the system remains polarized toward the stored pattern at all temperatures, meaning the overlap $m \neq 0$ for any $\beta > 0$. As the temperature decreases ($\beta \to \infty$), the

magnetization—i.e., the overlap with the pattern—monotonically increases and eventually converges to 1, indicating perfect alignment with the stored configuration in the zero-temperature limit.

If we now consider multiple patterns by generalizing the initial Hamiltonian (4), we arrive at the natural extension:

$$\mathcal{H} = -\sum_{\mu=1}^{P} \tau_\mu \sum_{i=1}^{N_\mathrm{v}} s_i w_{i\mu}, \tag{7}$$

which describes a system with $P > 1$ binary hidden nodes, each associated with a pattern $\boldsymbol{w}_\mu$. In the low-load regime $P \sim O(1)$ w.r.t. $N_\mathrm{v}$, the corresponding saddle-point equations for $m_\mu$ take the form:

$$m_\mu = \mathbb{E}_{\boldsymbol{w}}\left[w_\mu \tanh\left(\beta \sum_\nu w_\nu \operatorname{sign}(m_\nu)\right)\right], \quad \forall \mu = 1, \dots, P. \tag{8}$$

Here, $\{m_\mu\}_{\mu=1}^{P}$, with $m_\mu = N_\mathrm{v}^{-1} \sum_i w_{i\mu} s_i$, denotes the set of order parameters characterizing the overlaps between the system configuration and each stored pattern. The expectation $\mathbb{E}_{\boldsymbol{w}}$ appears as a consequence of the central limit theorem — with $N_\mathrm{v}^{-1} \sum_i^{N_\mathrm{v}} f(w_{i\mu}) \to \mathbb{E}_{\boldsymbol{w}} f(w_\mu)$ for a generic $f$— which justifies replacing empirical averages with their population counterparts in the thermodynamic limit. It is quite striking how the system differs from the Hopfield case due to the binary nature of the hidden nodes. In the Hopfield model, the activation of the hidden units (conditioned on the visible configuration) scales linearly with the local field $h_\mu = \sum_i w_{i\mu} s_i$, allowing for large deviations when the spins align with a stored pattern. In contrast, when the hidden nodes are binary, the system's response is bounded by the hyperbolic tangent, and the average activation of the hidden nodes (conditioned on the visible ones) is given by:

$$\langle \tau_\mu \rangle_{p(\tau_\mu|\boldsymbol{s})} = \tanh\left(\sum_i w_{i\mu} s_i\right). \tag{9}$$

It is still possible to modify the model (7) to recover behavior analogous to that of the Hopfield model. The key idea is to associate an *extensive* number of hidden nodes to each pattern, setting $N_\mathrm{h} = \alpha_\mathrm{H} N_\mathrm{v}$, with $\alpha_\mathrm{H} = \mathcal{O}(1)$ in the thermodynamic limit. The resulting Hamiltonian then takes the form:

$$\mathcal{H}\left(\boldsymbol{s}, \left\{\boldsymbol{\tau}^{(\mu)}\right\}_{\mu=1}^{P}\right) = -\frac{1}{N_\mathrm{v}} \sum_{\mu=1}^{P} \sum_{i=1}^{N_\mathrm{v}} s_i w_{i\mu} \sum_{a=1}^{N_\mathrm{h}} \tau_a^{(\mu)}, \tag{10}$$

where the hidden variables $\tau^{(\mu)}$ have been replicated $N_\mathrm{h}$ times for a single pattern $\mu$ and a factor $1/N_\mathrm{v}$ has been added to keep the Hamiltonian extensive. Introducing the order parameters as before, the partition function now reads

$$Z = \sum_{\boldsymbol{s}} \int \prod_\mu dm_\mu d\bar{m}_\mu \exp\left[N_\mathrm{h} \sum_\mu \log 2\cosh\left(\beta m_\mu\right) + \mathrm{i} N_\mathrm{v} \sum_\mu m_\mu \bar{m}_\mu - \mathrm{i} \sum_\mu \bar{m}_\mu \sum_i s_i w_{i\mu}\right]$$
$$= \int \prod_\mu dm_\mu \exp\left[\alpha_\mathrm{H} N_\mathrm{v} \sum_\mu \log 2\cosh\left(\beta m_\mu\right) + \mathrm{i} N_\mathrm{v} \sum_\mu m_\mu \bar{m}_\mu + \sum_i \log 2\cosh\left(\mathrm{i} \sum_\mu \bar{m}_\mu w_{i\mu}\right)\right], \tag{11}$$

which leads to the following saddle point equations

$$m_\mu = \mathbb{E}_{\boldsymbol{w}} w_\mu \tanh\left[\beta \alpha_\mathrm{H} \sum_\nu w_\nu \tanh\left(\beta m_\nu\right)\right] \quad \forall \mu = 1, \dots, P \tag{12}$$

A standard analysis of the above expression shows that the model exhibits a second-order phase transition at $\beta_\mathrm{c} = \sqrt{1/\alpha_\mathrm{H}}$, separating a paramagnetic phase from a ferromagnetic phase in which one pattern is expressed—that is, $m_\nu \neq 0$ for some $\nu$. Details of these derivations can be found in Appendix A. This behavior closely parallels the phase transition observed in the low-load Hopfield model and aligns with the results of [50], where a similar transition was found using a low-rank approximation of the coupling matrix.

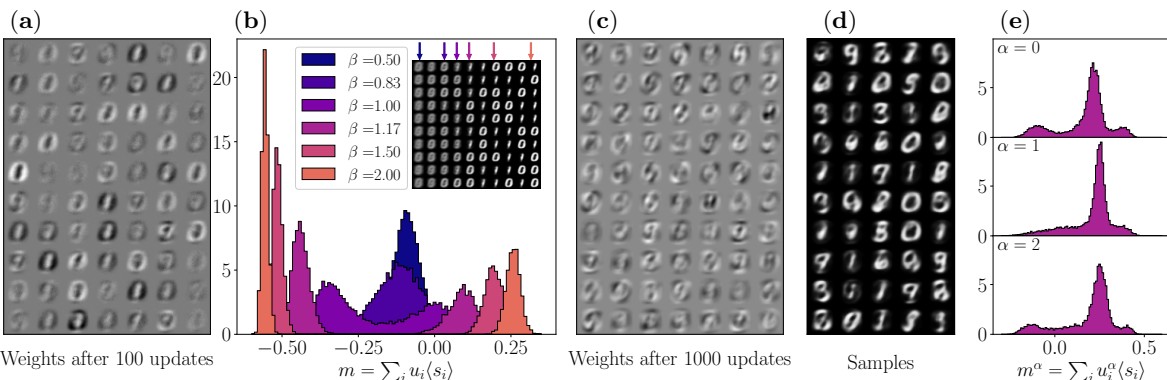

FIG. 2. **Illustration of the early learning dynamics of a binary-binary RBM trained on the binarized MNIST dataset.** (a) Columns of the weight matrix after 100 gradient updates, displaying a subset of 70 out of 100 hidden units plotted as $28 \times 28$ images. (b) Samples generated by the same model as in (a), evaluated at different effective inverse temperatures $\beta$. For each temperature, the histogram shows the magnetization projected along the first principal direction $\boldsymbol{u}$ of the weight matrix; insets display representative samples generated by the model. (c) Same as (a), but after 1000 gradient updates. The weight matrix now encodes a greater diversity of patterns. (d) Samples generated by the model after 1000 updates, as in (c), showing increased digit variety but limited intra-class variability. (e) Histogram of overlaps between the generated samples (after 1000 updates) and the first three principal components $\alpha$ of the weight matrix, indicating dominant alignment along a few learned directions.

However, there is a subtle but important distinction from the standard Hopfield case. Due to the binary nature of the hidden nodes, the effective activation function involves a $\tanh(\beta m_\mu)$ nonlinearity, which introduces a saturation effect at low temperatures. In contrast, in the Gaussian case, the response remains linear in $\beta m_\mu$, allowing for unbounded growth of the signal as temperature decreases. Nevertheless, when $\alpha_\mathrm{H} = 1$, we observe that the temperature dependence of the magnetization closely matches that of the Hopfield model, suggesting that the overall thermodynamic behavior remains consistent in this regime.

The main takeaway from this section is that, when using binary hidden nodes, achieving a recall phase requires compensating for the "weak" response of individual units by replicating them—i.e., using a large number of hidden nodes. But does this phenomenon persist when training an RBM on real data? To address this question, we perform the following experiment: we train a binary RBM (with $\{0, 1\}$ visible and hidden units) on the MNIST dataset, using $N_\mathrm{h} = 500$ hidden nodes—comparable to the input size $N_\mathrm{v} = 784$. To probe the initial stages of learning, we use a small learning rate and examine the system after 100 gradient updates.

Figure 2(a) displays the learned weights for 70 hidden units, reshaped as $28 \times 28$ images. Remarkably, many hidden nodes encode the same dominant feature, often resembling a digit "1" (or "0" if you focus on the exterior part). In panel (b), we analyze the same model under a rescaled energy landscape, introducing a temperature parameter $T = 1/\beta$. We track the magnetization projected along the principal direction of the weight matrix as a function of $T$, with generated samples shown in the inset. This temperature rescaling allows us to explore how the free energy landscape becomes biased toward memory-like states. As the temperature decreases (meaning $\beta$ increases), the system increasingly aligns with a learned pattern; at higher temperatures, it returns to a paramagnetic state. Notably, for $T \leq 1$, the model consistently generates samples strongly resembling the dominant learned features – a behavior analogous to that of the Hopfield model in the low-load regime.

When training the model for a longer period, specifically after 1000 updates, we observe that the RBM has learned a broader set of patterns. In Figure 2(c), the features associated with the weight matrix exhibit greater diversity compared to earlier stages, though many hidden units still encode similar patterns, indicating redundancy. Panel (d) shows that the RBM is now capable of generating different digits, although the variability within each digit class remains limited. Finally, in panel (e), we observe that a large fraction of the generated samples exhibit strong overlap with the first three principal components of the weight matrix, suggesting that generation is still dominated by a few prominent directions in feature space.

## IV.   HIGH-LOAD REGIME: INFLUENCE OF THE HIDDEN-UNIT PRIOR ON RETRIEVAL

In this section, we investigate the retrieval capabilities of RBMs in the presence of an extensive number of stored patterns, focusing on how different priors imposed on the hidden nodes influence performance. We analyze and compare these priors, discussing both the computational procedure and the inherent limitations of this approach. All analytical results are derived using replica theory from the statistical physics of disordered systems, within the replica-symmetric ansatz.

### A.   Binary hidden nodes without bias

We begin by considering the generalization of the Binary-Binary RBM to the high-load regime, where the number of stored patterns is extensive, $P = \alpha N_{\mathrm{v}}$ [58], with $\alpha = \mathcal{O}(1)$, and each pattern is associated with an extensive number of hidden nodes, as in Eq. (10). The Hamiltonian for this system is given by:

$$\mathcal{H}\left(\boldsymbol{s}, \{\boldsymbol{\tau}_u\}_{u=1}^{\alpha_{\mathrm{H}} N_{\mathrm{v}}}\right) = -\frac{1}{N_{\mathrm{v}}} \sum_{i=1}^{N_{\mathrm{v}}} \sum_{\mu=1}^{\alpha N_{\mathrm{v}}} s_i w_{i\mu} \sum_{u=1}^{\alpha_{\mathrm{H}} N_{\mathrm{v}}} \tau_{\mu,u}. \tag{13}$$

This construction leads to a model where the total number of hidden nodes scales quadratically with the number of visible units, i.e. $N_{\mathrm{h}} = \alpha \alpha_{\mathrm{H}} N_{\mathrm{v}}^2$. The analytical treatment of this model is very similar to the one of the Hopfield model with an extensive number of patterns [44]. First, we introduce the order parameters, namely the Mattis magnetization (the averaged overlap of the spin configuration over one randomly chosen pattern)[59], and the overlap between two replicas $a, b$ of the system

$$m = \frac{1}{N_{\mathrm{v}}} \sum_i w_{i1} s_i, \quad \text{and} \quad q_{ab} = \frac{1}{N_{\mathrm{v}}} \sum_i s_i^a s_i^b. \tag{14}$$

The Hamiltonian can then be decomposed into two contributions: the first captures the signal, namely the overlap with a reference pattern (typically the first, $\boldsymbol{w}_1$), while the second encompasses the remaining patterns, which act as quenched disorder and contribute to the system's intrinsic noise. Using standard techniques from the theory of disordered systems (see, e.g. [60]), we derive the saddle-point equations that dominate the measure in the thermodynamic limit $N_{\mathrm{v}} \to \infty$, shown below ( a complete derivation is provided in Appendix B):

$$m = \int \mathcal{D}z \tanh\left[\sqrt{\hat{q}}z + \alpha_{\mathrm{H}}\beta \tanh(\beta m)\right], \tag{15a}$$

$$q = \int \mathcal{D}z \tanh^2\left[\sqrt{\hat{q}}z + \alpha_{\mathrm{H}}\beta \tanh(\beta m)\right], \tag{15b}$$

$$\hat{q} = \alpha \frac{(\alpha_{\mathrm{H}}\beta^2)^2 q}{\Delta^2}, \tag{15c}$$

where $\Delta = 1 - \alpha_{\mathrm{H}}\beta^2(1 - q)$. The corresponding free energy is then given by

$$-\beta f = \alpha_{\mathrm{H}} \log 2 \cosh(\beta m) - \frac{\hat{q}(q-1)}{2} + \int \mathcal{D}z \log 2 \cosh(\sqrt{\hat{q}}z + \hat{m}) + m\hat{m} + \alpha \alpha_{\mathrm{H}}\beta^2 \frac{q}{2\Delta} - \frac{\alpha}{2} \log \Delta \tag{16}$$

Equations (15) can be solved numerically to construct the phase diagram for various values of $(\alpha_{\mathrm{H}}, \beta, \alpha)$. Before proceeding with this analysis, we highlight a noteworthy feature of the model. In the argument of the tanh function in Eqs. (15a)–(15b), the random field associated with the noise scales as $\alpha_{\mathrm{H}}\beta^2$, whereas the signal term (at large $\beta$) scales only linearly, as $\propto \alpha_{\mathrm{H}}\beta$. From this we can deduce that the system will not exhibit recall in the zero-temperature limit, since the effect of the noise will always dominate: this is clearly due to the fact that the hyperbolic tangent is bounded and therefore the return signal from the hidden nodes get clamped between $-1$ and $1$. On the contrary, in the Hopfield case the Gaussian response function will allow the signal to compete with the noise at zero temperature.

In Figure 3 (left), we show the boundary of the recall phase for different values of $\alpha_{\mathrm{H}}$ in the $(\alpha, T/\sqrt{\alpha_{\mathrm{H}}})$ plane. This boundary is computed as the maximum value of the pattern load $\alpha$ for

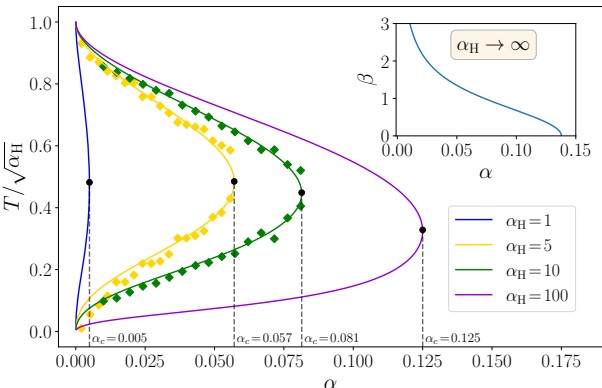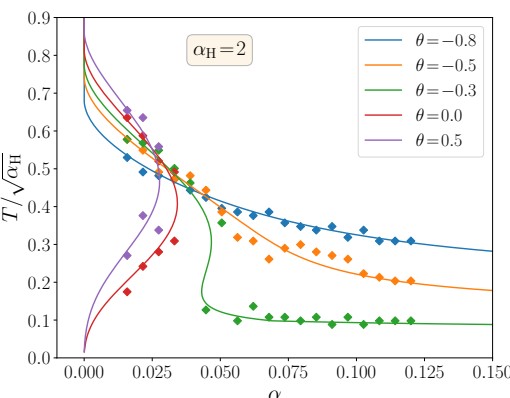

FIG. 3. **Left:** Recall phase of the binary-binary RBM with an extensive number of patterns, as defined in Eq. (13). Colored lines indicate the phase boundary of the (metastable) recall phase in the $(\alpha, T/\sqrt{\alpha_H})$ plane for various values of $\alpha_H$. Diamond-shaped scatter points represent numerical estimates obtained via Monte Carlo simulations for a system of size $N_v = 100$. **Inset:** Critical capacity computed in the limit $\alpha_H \to \infty$, and plotted in the plane $(\alpha, \beta)$. In this limit, the model recovers the Hopfield value $\alpha_c \approx 0.14$ when $\beta \to 0$. **Right:** Phase diagram of the model with sparse hidden activations, shown for $\alpha_H = 2$. As the bias $\theta$ increases, the model approaches the standard binary case; in contrast, decreasing $\theta$ silences most hidden nodes by favoring $\tau = 0$, effectively reducing their contribution. Interestingly, increasing sparsity (i.e., larger $\theta$) enhances the model's capacity, as more selective hidden activation reduces interference between patterns. This trend is corroborated by numerical simulations (in colored diamonds).

which a ferromagnetic solution (aligned with one pattern) exists as a local minimum of the free energy—corresponding to the spinodal point of the retrieval solution. The recall phase is metastable and exhibits a non-monotonic behavior with temperature: it reaches a maximum capacity at an intermediate $T$, then bends downward and vanishes as $T \to 0$ as expected according to previous discussion.

A key observation is that the critical capacity, here defined as $\alpha_c = \max_T \alpha(T)$, remains low unless $\alpha_H$ is extremely large. This indicates that the model is not well-suited for associative memory tasks in practice. For instance, even with a hidden-node density as high as $\alpha_H \sim 100$, the critical capacity barely matches that of the standard Hopfield model. The scatter points in the figure represent numerical estimates of the phase boundary obtained via Monte Carlo simulations for a system of size $N_v = 100$, which already represents a very good agreement with theoretical predictions. Further increasing the system size $N_v$ is computationally expensive, as the total number of hidden nodes scales quadratically with it.

As previously discussed, the zero-temperature limit of this model invariably leads to a spin-glass phase. However, we can theoretically explore an alternative asymptotic regime in which the density of hidden nodes, $\alpha_H$, is sent to infinity. Indeed, from Eq. (15a), we observe that the argument of the hyperbolic tangent scales proportionally with $\alpha_H$. This suggests that, analogously to the zero-temperature limit in the Hopfield model, we can consider the $\alpha_H \to \infty$ limit. In this regime, the saddle-point equations reduce to a single self-consistent equation of the form:

$$\beta\sqrt{2\alpha}x = \tanh\left(\beta \mathrm{erf}\,(x)\right) - \frac{2}{\sqrt{\pi}}x\beta e^{-x^2} \quad \text{with} \quad x = \frac{\tanh(\beta m)(1 - \beta^2 C)}{\beta\sqrt{2\alpha}}, \tag{17}$$

where $C = \lim_{\alpha_H \to \infty} \alpha_H(1 - q)$ remains finite. This scaling allows for a numerical determination of the critical capacity in the high-$\alpha_H$ regime. In the inset of the left panel of Figure 3, we show how the critical capacity evolves as a function of temperature in this limit, gradually approaching the Hopfield model value when $\beta \to 0$ as expected.

## V. FROM BINARY HIDDEN NODES TO RECTIFIED LINEAR UNITS

We have seen that replacing Gaussian hidden variables with binary $\tau = \pm 1$ ones significantly reduces the model's critical capacity and suppresses the ordered phase at low temperatures. However, in the limit of infinite hidden-node density, the model recovers the critical capacity of the classical Hopfield model. This is expected, as the sum of many independent binary variables converges to a Gaussian

distribution by the central limit theorem. Still, increasing the number of hidden nodes also amplifies the cumulative noise from other patterns, which limits further improvements in capacity. From this perspective, it is not surprising that the model's performance plateaus. An alternative approach, based on truncated Gaussian priors (equivalent to ReLU activations [61]), has been shown to yield a better phase diagram at zero temperature—surpassing even the Hopfield model in terms of critical capacity [48].

In this section, we show how to improve the performance of RBMs with binary hidden units by introducing a key modification: a local bias on the hidden nodes. By tuning this bias to control the scaling of the number of activated hidden units, we obtain a well-defined thermodynamic limit that outperforms the previous binary case without bias. We then extend this analysis to RBMs with truncated Gaussian hidden units, demonstrating how a properly chosen local field can suppress the spin-glass phase and further enhance retrieval capabilities.

### A. Ternary hidden nodes with adjusted biases

The intuition behind the poor critical capacity of the RBM with an extensive number of binary $\tau = \pm 1$ hidden units per pattern is that the activity coming from other patterns tends to destroy the ferromagnetic order. Given the large number of hidden units per pattern, this degradation is perhaps unsurprising. To address this issue, we introduce a modified model. Specifically, we consider hidden units that take three possible states, $\tau \in \{0, \pm 1\}$, allowing inactive nodes to be silenced via a local bias or magnetic field. The $\pm 1$ states are retained to preserve the symmetry of the Hamiltonian. With these ternary hidden variables, we define the following Hamiltonian:

$$\mathcal{H} = -\frac{1}{N_{\rm v}} \sum_{i,\mu} s_i w_{i\mu} \sum_u \tau_{\mu,u} - \sum_{u,\mu} \theta_u \left(1 - \delta_{\tau_{\mu,u},0}\right). \tag{18}$$

where each local bias $\theta_u$ can push (or inhibit) the corresponding hidden node $\tau_{\mu,u}$. This formulation enables us to analyze how both the bias and the introduction of three-state hidden nodes ($\tau \in \{0, \pm 1\}$) affect the free energy of the system. The details are postponed to the Appendix C. The obtained expression for the free energy is

$$-\beta f = m\hat{m} - \frac{\hat{q}(q-1)}{2} + \alpha \alpha_{\rm H} \sigma_0(\beta\theta)\beta^2 \frac{q}{2\Delta} - \frac{\alpha}{2} \log \Delta \tag{19}$$

$$+ \alpha_{\rm H} N_{\rm v}^{-1} \sum_u \log \left[1 + 2e^{\beta\theta_u} \cosh(\beta m)\right] + \int \mathcal{D}z \log 2 \cosh\left(\sqrt{\hat{q}}z + \hat{m}\right), \tag{20}$$

where

$$\Delta = 1 - \alpha_{\rm H} \sigma_0(\beta\theta)\beta^2(1-q) \tag{21}$$

$$\sigma_0(\beta\theta) = \lim_{N_{\rm v} \to \infty} N_{\rm v}^{-1} \sum_u {\rm sig}(\beta\theta_u + \log 2) \tag{22}$$

and sig is the sigmoid function. In the following, we consider a constant bias $\theta_u = \theta$ for simplicity. However, it is worth noting that a similar construction (replicating each feature while incrementally adjusting its bias) was used in the introduction of ReLU activations for RBMs in [61]. Under the assumption of a constant bias, we obtain the following expressions for the order parameters:

$$m = \int \mathcal{D}z \tanh(\sqrt{\hat{q}}z + \hat{m}) \tag{23a}$$

$$\hat{m} = \alpha_{\rm H}\beta {\rm sig}\left[\beta\theta + \log 2 \cosh(\beta m)\right] \tanh(\beta m), \tag{23b}$$

$$q = \int \mathcal{D}z \tanh^2(\sqrt{\hat{q}}z + \hat{m}) \tag{23c}$$

$$\hat{q} = \alpha \frac{q(\alpha_{\rm H}{\rm sig}(\beta\theta + \log 2)\beta^2)^2}{\left[1 - \alpha_{\rm H}{\rm sig}(\beta\theta + \log 2)\beta^2(1-q)\right]^2}. \tag{23d}$$

The bias can thus be tuned to suppress the noise associated with the order parameter $\hat{q}$. In the signal term, within the large $\beta$ (low-temperature) regime, the argument of the sigmoid function becomes

$\beta(\theta + |m|)$. This implies that negative bias values with $-1 < \theta < 0$ still allow a non-zero signal even at zero temperature, while the noise contribution can be effectively suppressed at sufficiently low temperatures because the term will be typically dominated by $\sim \text{sig}(\beta\theta)$. Indeed, we find that for any $\theta$ in the range $-1 < \theta < 0$ and $T \ll 1$, a retrieval phase persists. In the right panel of Figure 3, we illustrate the recall behavior of the model for several values of $\theta$. It is clear that small negative values of $\theta$ enhance the retrieval properties at low temperature by silencing the hidden nodes that do not receive a strong signal, suggesting that this mechanism is very efficient to improve the associative memory of these models.

## B. Truncated Gaussian hidden nodes

Rectified Linear Units (ReLUs) were first introduced in the context of RBMs by Hinton [61]. The recall capacity of RBMs with ReLU or Truncated Gaussian hidden nodes has since been investigated in the zero-temperature limit [48, 49]. Here, we extend this analysis by providing a full characterization of the recall properties in the temperature – pattern-density plane. The resulting phase diagram offers valuable insights into the behavior of RBMs during learning. In particular, since RBMs are typically trained at finite temperature (i.e., not in the zero-temperature or infinite-coupling limit), it is essential to understand their thermodynamics in this regime. In the first part, we detail the analytical derivation of the phase diagram. In the second, we show how hidden biases can mitigate the spin-glass phase—an obstacle to successful recall at low temperatures—and illustrate this effect with numerical simulations.

### 1. Associative memory using Tuncated Gaussian hidden nodes with no bias

To introduce ReLU-like activations, we consider the starting Hamiltonian of the RBM given in (1) with the following prior distribution imposed on the hidden nodes:

$$p(\tau) \propto \theta(\tau) \exp\left(-\lambda \frac{\tau^2}{2}\right). \tag{24}$$

corresponding to the hidden potential:

$$\mathcal{U}^{(\text{h})}(\tau) = \begin{cases} -\frac{\lambda}{2}\tau^2 & \tau > 0 \\ \infty & \text{otherwise} \end{cases} \tag{25}$$

Here, the variable $\tau$ is constrained to take only non-negative values, and the parameter $\lambda$ controls the variance of the distribution. This choice of prior is particularly useful as it suppresses negative contributions to the hidden activations, effectively mimicking the behavior of rectified linear units. Once again, the mean-field theory corresponding to this choice of the prior can be derived using standard techniques from the replica method (detailed in Appendices D and E). This leads to the following expression for the free energy:

$$f = m\hat{m} + \frac{\alpha\beta}{2}\hat{q}(1-q) - \frac{m^2}{2\lambda}\Theta(m) + \frac{\alpha}{2\beta}\log\Delta - \frac{\alpha q}{2\Delta} - \frac{\alpha}{\beta}\int Dz \log H(az)$$
$$- \frac{1}{\beta}\mathbb{E}_w \int Dz \log 2\cosh\left[\beta\hat{m} + \beta\sqrt{\alpha\hat{q}}z\right] \tag{26}$$

with $a = \sqrt{\frac{\beta q}{\Delta}}$ and $\Delta = \lambda - \beta(1-q)$. The corresponding saddle point equations are

$$m = \int Dz \tanh\left[\beta\hat{m} + \beta\sqrt{\alpha\hat{q}}z\right], \tag{27a}$$

$$q = \int Dz \tanh^2\left[\beta\hat{m} + \beta\sqrt{\alpha\hat{q}}z\right], \tag{27b}$$

$$\hat{m} = \max\left(0, \frac{m}{\lambda}\right), \tag{27c}$$

$$\hat{q} = \frac{1}{\Delta^2}(q + \eta^2) + \frac{1}{2\pi\beta\Delta}\frac{(\lambda - \beta)}{\Delta + \beta q}\int Dz \frac{e^{-(az)^2}}{H^2(az)}, \tag{27d}$$

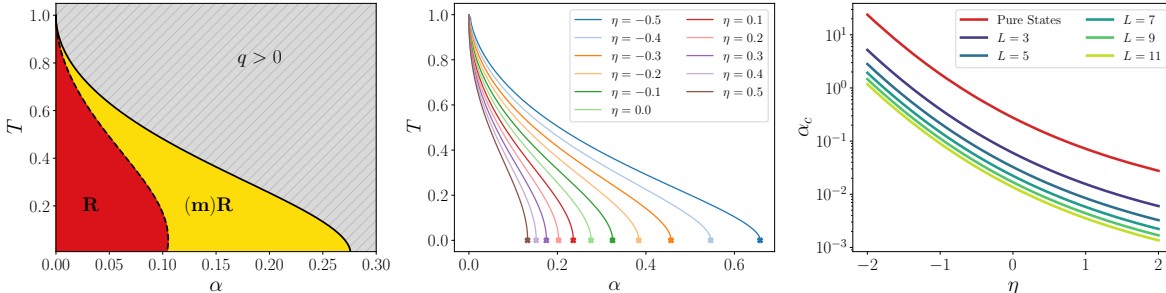

FIG. 4. **Left:** Phase diagram of the ReLU RBM as a function of temperature. The retrieval (R) and metastable-recall (m)R phases are significantly larger than in the Hopfield model, indicating enhanced memory capacity. A Spin glass phase is also present, yet the line is more difficult to identify analytically since that the overlap parameter $q$ is always non-zero due to the non-symmetric nature of the hidden variables. **Center:** Transition lines between the (m)R and spin-glass phases are shown for various values of the hidden bias $\eta$. As $\eta$ becomes more and more negative, the phase boundary shifts, enlarging the retrieval region. The light green curve corresponds to the case $\eta = 0$ (and thus to the boundary in the left panel). **Right:** Critical capacity of the model as a function of the hidden bias $\eta$, shown for the retrieval of $L$ memory states.

where we defined the function

$$H\left(x\right) = \frac{1}{2}\mathrm{erfc}\left(\frac{x}{\sqrt{2}}\right).$$ (28)

While the zero-temperature limit of such a system has already been discussed in [48, 49], we are here interested in the full phase diagram in temperature, and the effect of a magnetic field (in the next subsection). First, let us discuss the phase diagram shown in the leftmost panel of Figure 4. The recall area is qualitatively similar to the Hopfield case, albeit it spans a larger portion of the plane $\alpha - T$: indeed, the critical capacity is higher (about twice the Hopfield value). Still, the main issue is that the spin glass phase is present (grey area in Figure 4-left). There is another difference w.r.t. to the Hopfield case: no second-order phase transition exists between a paramagnetic phase with both $m, q = 0$ and a pure spin-glass phase with $m = 0, q > 0$. In other terms, in the grey area the RS overlap is always positive (although decreasing with increasing temperature). This behavior seems to be related to the fact that hidden nodes are defined only in half the real axis, effectively constraining the volume of configurations in the visible space. Although one could in principle study the nature of such a spin-glass phase, e.g. by looking at the stability of the RS solution to check where a transition to RSB occurs (e.g. using the de Almeida&Thouless formalism [62]), here we only focus on retrieval capabilities of the model, leaving this issue for future investigations.

### 2. Associative memory using Truncated Gaussian hidden nodes with a hidden bias

We can straightforwardly introduce a bias term on the hidden nodes. This modification makes it more difficult to activate hidden units that are not aligned with a specific pattern, effectively enhancing selectivity. The resulting Hamiltonian takes the form:

$$\mathcal{H}[\boldsymbol{s}, \boldsymbol{\tau}] = -\sum_{i,\mu} s_i w_{i\mu}\tau_\mu - \eta\sum_\mu \tau_\mu - \sum_\mu \mathcal{U}^{(\mathrm{h})}\left(\tau_\mu\right).$$ (29)

The presence of this field implies the following modification in the conjugate parameter $\hat{q}$ of the replica overlap

$$\hat{q} = \frac{1}{\Delta^2}\left(q + \eta^2\right) + \sqrt{\frac{2}{\pi\beta\Delta}}\left[\frac{\eta}{\Delta + \beta q}\right]\int Dz \frac{e^{-\frac{1}{2}(az+b)^2}}{H\left(az+b\right)} + \frac{1}{2\pi\beta\Delta}\frac{\lambda - \beta}{\Delta + \beta q}\int Dz \frac{e^{-(az+b)^2}}{H^2\left(az+b\right)}$$ (30)

where $b = -\eta\sqrt{\beta/\Delta}$. The primary effect of the magnetic field is to stabilize the retrieval of patterns when it takes negative values, as illustrated in the center and right panels of Figure 4. In the right panel, we observe that the critical capacity—inferred at $T = 0$ for various numbers of recalled patterns $L$—systematically increases as the field becomes more negative. Conversely, a positive field tends to

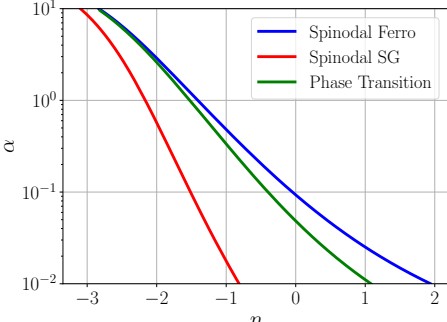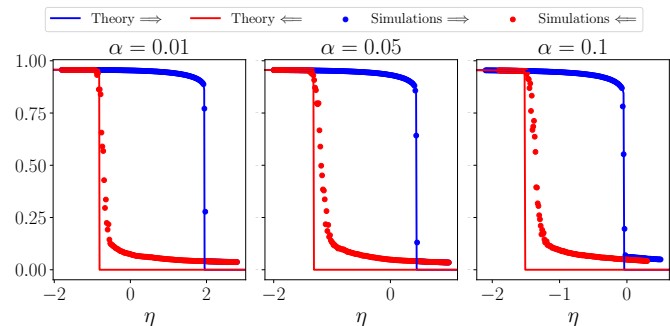

FIG. 5. **Left:** Critical lines in the Binary-ReLU RBM with an external field, plotted in the plane $(\eta, \alpha)$. The blue line represents the spinodal of the Ferromagnetic solution, at the right of which no retrieval state is present. The red line represents the spinodal of the spin-glass solution, at the left of which no such fixed point exists. Finally, the green curve represent the (1st) order phase transition line between the two equilibrium states. **Right:** The overlap between a given pattern and the sampled configuration. In blue, doing a annealing from $\eta = -2$ to higher valuer, in orange an annealing starting from $\eta = -0.5$ toward higher values. We can clearly identify the two spinodal points. Here $N_{\rm v} = 10000$, $\alpha = 0.5$ and at each annealing steps we perform $t = 10^4$ MC steps.

suppress pattern recall. Notably, even for moderate values of the hidden bias, the (metastable) recall region significantly expands toward higher values of $\alpha$.

Even more interestingly, we investigate how the spin-glass phase is affected by the hidden bias. We find that there exists a critical value of the field $\eta$, beyond which the spin-glass phase disappears entirely. In this regime, the system is able to recall a pattern without being trapped in a spin-glass state. In the left panel of Figure 5, we plot the critical line $\alpha(\eta_{\rm c})$ in the $\eta - \alpha$ plane for a fixed temperature. To validate these theoretical predictions, we perform numerical experiments using a binary-ReLU RBM with binary patterns and a global hidden bias. We run Monte Carlo simulations with $N_{\rm v} = 10^4$ visible units at various $\eta$ and $\alpha$ and at fixed temperature $T = 0.5$. Starting from random initial conditions, we perform $t = 10^4$ Monte Carlo sweeps and measure the maximal overlap between the spin configuration and the stored patterns. This quantity is averaged over $N_{\rm R} = 500$ disorder realizations. The right panel of Figure 5 shows that the average overlap increases sharply when the bias $\eta$ falls below the critical value $\eta_{\rm c}(\alpha, T)$, confirming the disappearance of the spin-glass phase and the onset of successful pattern retrieval.

*A final remark on the ReLU potential*, It should be noted, however, that the Truncated Gaussian prior excludes the possibility to have a finite fraction of totally de-activated hidden units. In order to account for it, the correct prior for this choice of ReLU hidden units should be a *rectified* Gaussian distribution rather than a *truncated* one. The first one can be simply expressed as a mixture between the second one and a delta peak in 0, whose weight precisely accounts for the missing probability mass in the interval $(-\infty, 0)$. Implementing such a choice results into a slight modified mean-field theory, for which however no notable difference has been found w.r.t. the truncated case.

## VI. CONCLUSION

In this work, we analyzed the retrieval properties of bipartite energy-based models, focusing on Restricted Boltzmann Machines (RBMs) with binary visible units and different architectural choices for the hidden layer. We showed that, unlike their Gaussian counterparts or the classical Hopfield model, RBMs with binary visible and hidden units suffer from a markedly reduced retrieval capacity in the high-load regime. This limitation stems from the saturation of the binary activation function and increased interference among competing patterns, which collectively hinder the emergence of robust memory states.

We demonstrated that this limitation can be partially alleviated by introducing redundant hidden units per pattern or by applying local biases to regulate their activation. However, such modifications do not fully restore the associative memory capacity seen in the Hopfield model. A more effective strategy involves relaxing the binary constraint: using ternary or ReLU-like hidden variables dramatically enhances performance. In particular, the introduction of tunable local biases allows the model

to suppress spurious activations, stabilize recall states, and avoid the onset of spin-glass phases—even at finite temperatures.

Our theoretical predictions, grounded in replica computations and validated by finite-size Monte Carlo simulations, highlight the critical role of architectural choices—especially the hidden unit prior—in shaping the expressive power and functional behavior of generative neural models. These findings suggest that the capacity for associative memory is not the sole or even dominant mechanism underlying the generative performance of a model. In fact, RBMs with binary hidden units can substantially outperform Hopfield networks in terms of generation quality, despite their weaker capacity for memorization. This observation underscores the need to better understand the interplay between memorization, generalization, data structure and architectural design in unsupervised learning systems.

### ACKNOWLEDGEMENTS

Authors acknowledge financial support by the Comunidad de Madrid and the Complutense University of Madrid through the Atracción de Talento program (Refs. 2019-T1/TIC-13298 & Refs. 2023- 5A/TIC-28934), the project PID2021-125506NA-I00 financed by the "Ministerio de Economía y Competitividad, Agencia Estatal de Investigación" (MICIU/AEI/10.13039/501100011033), the Fondo Europeo de Desarrollo Regional (FEDER, UE).

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

**Appendix A: Derivation of the free energy of the low-load cases**

We provide the analytical details of the computation for the low-load case with binary hidden nodes, i.e. the models discussed in Section III.

**1. Intensive number of hidden nodes**

We start by the simplest model introduced in Eq. (7) with $P \sim O(1)$ patterns. The case $P = 1$ corresponding to the Hamiltonian (4) can be derived afterwards by simply setting $P = 1$. Notice first that the Hamiltonian (7) can be rewritten as a function of a set of Mattis magnetizations, one for each pattern:

$$\mathcal{H} = -\sum_{\mu=1}^{P} \tau_\mu \sum_{i=1}^{N_v} s_i w_{i\mu} = -N_v \sum_{\mu=1}^{P} \tau_\mu m_\mu(\boldsymbol{s}) \quad \text{where} \quad m_\mu(\boldsymbol{s}) = \frac{1}{N_v} \sum_i w_{i\mu} s_i \tag{A.1}$$

The partition function can be easily computed using a set of Dirac-delta distributions to impose the Mattis magnetization over all the patterns:

$$1 = \prod_{\mu=1}^{P} \int dm_\mu \delta\left(m_\mu - \frac{1}{N_v}\sum_i w_{i\mu}s_i\right) = \prod_{\mu=1}^{P} \int d\hat{m}_\mu \exp\left[iN_v m_\mu \hat{m}_\mu - i\hat{m}_\mu \sum_i w_{i\mu}s_i\right]. \tag{A.2}$$

Using the above expression and the shorthand notations $d\boldsymbol{m} = \prod_\mu dm_\mu$ (and similarly for $\hat{\boldsymbol{m}}$) we can write the partition function of Eq. (A.1) as

$$Z = \sum_{\boldsymbol{s},\boldsymbol{\tau}} \exp\left(\beta \sum_{\mu=1}^{P} \tau_\mu \sum_{i=1}^{N_v} s_i w_{i\mu}\right), \tag{A.3}$$

$$= \sum_{\boldsymbol{s}} \exp\left[\sum_\mu \log 2\cosh\left(\beta \sum_i w_{i\mu}s_i\right)\right], \tag{A.4}$$

$$= \sum_{\boldsymbol{s}} \int d\boldsymbol{m}d\hat{\boldsymbol{m}} \, \exp\left[\sum_\mu \log\cosh\left(\beta N_v m_\mu\right) + iN_v \boldsymbol{m}\cdot\hat{\boldsymbol{m}} - i\sum_\mu \hat{m}_\mu \sum_i s_i w_{i\mu}\right], \tag{A.5}$$

$$= \int d\boldsymbol{m}d\hat{\boldsymbol{m}} \, \exp\left[\sum_\mu \log\cosh\left(\beta N_v m_\mu\right) - iN_v \boldsymbol{m}\cdot\hat{\boldsymbol{m}} - \sum_i \log 2\cosh\left(i\sum_\mu w_{i\mu}\hat{m}_\mu\right)\right]. \tag{A.6}$$

Now we can find the saddle points by derivating the exponent of (A.6) w.r.t. $\{(m_\mu, \hat{m}_\mu)\}_{\mu=1}^{P}$:

$$m_\mu = \frac{1}{N_v}\sum_i w_{i\mu} \tanh\left(i\sum_\nu w_{i\nu}\hat{m}_\nu\right) \tag{A.7}$$

$$i\hat{m}_\mu = \beta \tanh\left(\beta N_v m_\mu\right) \tag{A.8}$$

In the limit $N_v \to \infty$, by using the property $\tanh\left(\beta N_v m_\mu\right) \to \text{sign}\left(m_\mu\right)$ and by replacing the empirical average w.r.t. one realization of the set of stored patterns $\boldsymbol{w}$ with the expectation value w.r.t. their distribution, we get:

$$m_\mu = \mathbb{E}_{\boldsymbol{w}}\left[w_\mu \tanh\left(i\sum_\nu w_\nu \hat{m}_\nu\right)\right] \tag{A.9}$$

$$i\hat{m}_\mu = \beta\text{sign}\left(m_\mu\right) \tag{A.10}$$

The combination of the two above equations gives Eq. (8).

### 2. Extensive number of hidden nodes

In order to correct the behavior of the previous model, an extensive number of hidden nodes $N_{\text{h}} = 1, \ldots, \alpha_{\text{H}} N_{\text{v}}$, leading to the Hamiltonian defined in Eq. (10) and re-written here for convenience:

$$\mathcal{H}\left(\boldsymbol{s}, \left\{\boldsymbol{\tau}^{(\mu)}\right\}_{\mu=1}^{P}\right) = -\frac{1}{N_{\text{v}}} \sum_{\mu=1}^{P} \sum_{i=1}^{N_{\text{v}}} s_i w_{i\mu} \sum_{a=1}^{N_{\text{h}}} \tau_a^{(\mu)}, \tag{A.11}$$

With this modification, introducing two sets of order parameters as

$$m_\mu^{(\text{v})} = \frac{1}{N_{\text{v}}} \sum_i s_i w_{i\mu} \qquad \text{and} \qquad m_\mu^{(\text{h})} = \frac{1}{N_{\text{h}}} \sum_a \tau_a^{(\mu)}, \tag{A.12}$$

for $\mu = 1, \ldots, P$, the partition function can be obtained by introducing the order parameters as in the previous section:

$$Z = \sum_{\{\boldsymbol{s}\},\{\boldsymbol{\tau}\}} \exp\left[\frac{\beta}{N_{\text{v}}} \sum_{\mu=1}^{P} \sum_{i=1}^{N_{\text{v}}} s_i w_{i\mu} \sum_{a=1}^{N_{\text{h}}} \tau_a^{(\mu)}\right] \tag{A.13}$$

$$= \sum_{\{\boldsymbol{s}\},\{\boldsymbol{\tau}\}} \int \mathcal{D}[\boldsymbol{m}] e^{\beta N_{\text{h}} \sum_\mu m_\mu^{(\text{v})} m_\mu^{(\text{h})}} \prod_\mu \delta(N_{\text{v}} m_\mu^{(\text{v})} - \sum_i s_i w_{i\mu}) \prod_\mu \delta(N_{\text{h}} m_\mu^{(\text{h})} - \sum_a \tau_a^{(\mu)}) \tag{A.14}$$

$$= \sum_{\{\boldsymbol{s}\},\{\boldsymbol{\tau}\}} \int \mathcal{D}[\boldsymbol{m}] \exp\left[\beta N_{\text{h}} \sum_\mu m_\mu^{(\text{v})} m_\mu^{(\text{h})} - iN_{\text{v}} \sum_\mu \hat{m}_\mu^{(\text{v})} m_\mu^{(\text{v})} + i\sum_\mu \hat{m}_\mu^{(\text{v})} \sum_i s_i w_{i\mu}\right.$$

$$\left. -iN_{\text{h}} \sum_\mu \hat{m}_\mu^{(\text{h})} m_\mu^{(\text{h})} + i\sum_\mu \hat{m}_\mu^{(\text{h})} \sum_a \tau_a^{(\mu)}\right] \tag{A.15}$$

$$= \int \mathcal{D}[\boldsymbol{m}] \exp\left[\beta N_{\text{h}} \sum_\mu m_\mu^{(\text{v})} m_\mu^{(\text{h})} - iN_{\text{v}} \sum_\mu \hat{m}_\mu^{(\text{v})} m_\mu^{(\text{v})} - iN_{\text{h}} \sum_\mu \hat{m}_\mu^{(\text{h})} m_\mu^{(\text{h})}\right. \tag{A.16}$$

$$\left. + \sum_i \log 2\cosh\left(i\sum_\mu \hat{m}_\mu^{(\text{v})} w_{i\mu}\right) + N_{\text{h}} \sum_\mu \log 2\cosh\left(i\hat{m}_\mu^{(\text{h})}\right)\right] \tag{A.17}$$

where we used the short-hand notation $\mathcal{D}[\boldsymbol{m}] = \mathrm{d}\boldsymbol{m}^{(\text{v})} \mathrm{d}\boldsymbol{m}^{(\text{h})}$. Taking the saddle point when $N_{\text{v}} \to \infty$, we arrive to the following free energy

$$-\beta f = \beta \alpha_{\text{H}} \sum_\mu m_\mu^{(\text{v})} m_\mu^{(\text{h})} - i\sum_\mu \hat{m}_\mu^{(\text{v})} m_\mu^{(\text{v})} - i\alpha_{\text{H}} \sum_\mu \hat{m}_\mu^{(\text{h})} m_\mu^{(\text{h})} + \tag{A.18}$$

$$+ \mathbb{E}_{\boldsymbol{w}} \log 2\cosh\left(i\sum_\mu \hat{m}_\mu^{(\text{v})} w_\mu\right) + \alpha_{\text{H}} \sum_\mu \log 2\cosh(i\hat{m}_\mu^{(\text{h})}) \tag{A.19}$$

The relevant equations for the order parameters $m_\mu^{(\text{v})}, m_\mu^{(\text{h})}$ (obtained after substituting the conjugate ones) are given by

$$m_\mu^{(\text{v})} = \mathbb{E}_{\boldsymbol{w}}\left[w_\mu \tanh\left(\beta \alpha_{\text{H}} \sum_\nu m_\nu^{(\text{h})} w_\nu\right)\right], \tag{A.20}$$

$$m_\mu^{(\text{h})} = \tanh\left(\beta m_\mu^{(\text{v})}\right). \tag{A.21}$$

Finally, substituting (A.21) into (A.20) we get to Eq. (12) in the main text. As said before, investigating for the recall of one representative pattern, a second-order phase transition appears $\beta^2 \alpha_{\text{H}} = 1$, hence $\beta_{\text{c}} = \sqrt{1/\alpha_{\text{H}}}$.

### 3. Low-load and Rectified Linear units

For the sake of completeness let's investigate what happens when using Rectified Linear Units, which correspond by taking a truncated Gaussian for the prior on the hidden nodes. In this context, we go

back to the classical setting where one hidden node encodes for one pattern. The hidden units $\tau_a$ can only be positive. Therefore the prior is given by

$$p(\tau) \propto \exp\left(-\frac{\tau^2}{2\sigma_\tau^2}\right) \quad \text{for } \tau \in \mathbb{R}^+. \tag{A.22}$$

The method to compute the partition function is very similar to the case of the Hopfield model, with the restriction that the hidden node is constrained to be positive.

$$Z = \int d\hat{\boldsymbol{m}} d\boldsymbol{m} d\boldsymbol{\tau} \exp\left(-\sum_a \frac{\tau_a^2}{2\sigma_\tau^2} + N_{\mathrm{v}}\beta \sum_a \tau_a m_a - iN \sum_a \hat{m}_a m_a + \sum_i \log \cosh\left[i \sum_a \hat{m}_a w_{ia}\right]\right). \tag{A.23}$$

Eliminating the conjugate variables $\hat{\boldsymbol{m}}$, we obtain

$$-\beta f(\boldsymbol{m}) = -\sum_a \frac{\beta^2 \sigma_\tau^2}{2} \max(0, m_a^2) + \left\langle \log \cosh\left[\beta^2 \sigma_\tau^2 \sum_a w_a \max(0, m_a)\right]\right\rangle. \tag{A.24}$$

Taking the saddle point, we obtain the following mean field equation in the case of discrete weights:

$$m_a = \left\langle w_a \tanh(\beta \sum_b \tau_b w_b)\right\rangle,$$

$$\tau_a = \sigma_\tau^2 \beta \max(0, m_a).$$

We see that in the case that the magnetization is positive the behavior on $\tau$ is linear in the magnetization. In the end, taking into account the constraint on $\tau \geq 0$, we get

$$m_a = \left\langle w_a \tanh\left(\beta^2 \sigma_\tau^2 \sum_b \max(0, m_b)\right)\right\rangle. \tag{A.25}$$

The second order phase transition can be easily computed by considering the recall of only one pattern (considering binary $\pm 1$ random i.i.d. patterns) and taking small values of $m_a$. We obtain that

$$\beta_{\mathrm{c}} = \sigma_\tau. \tag{A.26}$$

### Appendix B: Derivation of the free energy of the high-load binary-binary RBM with repetead hidden nodes

We start from the Hamiltonian defined in Eq. (13), and consider the replicated partition function with $n$ replicas with same quenched disorder (here given by the patterns' distribution):

$$Z^n = \sum_{\left\{s_i^{(a)}, \tau_{\mu,u}^{(a)}\right\}} \exp\left[\frac{\beta}{N_{\mathrm{v}}} \sum_{a=1}^n \sum_{i=1}^{N_{\mathrm{v}}} s_i^{(a)} w_{i1} \sum_{u=1}^{\alpha_{\mathrm{H}} N_{\mathrm{v}}} \tau_{1,u}^{(a)} + \frac{\beta}{N_{\mathrm{v}}} \sum_a \sum_i s_i^{(a)} \sum_{\mu>1}^P w_{i\mu} \sum_u \tau_{\mu,u}^{(a)}\right], \tag{B.1}$$

where we have already separated the signal contribution from a —just one, for simplicity— representative pattern $\boldsymbol{w}_1$. The average over the disorder reads:

$$\left\langle e^{\frac{\beta}{N_{\mathrm{v}}} \sum_a \sum_i s_i^{(a)} \sum_{\mu>1}^P w_{i\mu} \sum_u \tau_{\mu,u}^{(a)}}\right\rangle_{\boldsymbol{w}} = \prod_{i,\mu>1} \left\langle e^{\frac{\beta}{N_{\mathrm{v}}} w_{i\mu} \sum_a s_i^{(a)} \sum_u \tau_{\mu,u}^{(a)}}\right\rangle$$

$$= \prod_{i,\mu>1} 2\cosh\left[\frac{\beta}{N_{\mathrm{v}}} \sum_a s_i^{(a)} \sum_u \tau_{\mu,u}^{(a)}\right]$$

$$\approx \exp\left[\frac{1}{2} \frac{\beta^2}{N_{\mathrm{v}}^2} \sum_{i,\mu>1} \left(\sum_a s_i^{(a)} \sum_u \tau_{\mu,u}^{(a)}\right)^2\right]. \tag{B.2}$$

Substituting this result in the above expression and introducing the order parameters (Introducing the Mattis magnetizations $m^a = \frac{1}{N_v}\sum_i s_i^{(a)} w_{i,1}$ and the 2-replica overlap $q_{ab} = \frac{1}{N_v}\sum_i s_i^{(a)} s_i^{(b)}$ we get, after some straightforward manipulations:

$$\langle Z^n \rangle = \int \prod_a (dm^a d\hat{m}^a) \prod_{a<b} (dq_{ab}\hat{q}_{ab}) \, e^{iN_v \sum_a m^a \hat{m}^a + iN_v \sum_{a<b} q_{ab}\hat{q}_{ab}}$$

$$\times \prod_i \left[ \sum_{s_i^{(a)}} e^{-i\sum_a \hat{m}^a s_i^{(a)} w_{i,1} - i\sum_{a<b} \hat{q}_{ab} s_i^{(a)} s_i^{(b)}} \right] \times \prod_{u,a} \sum_{\tau_{1,u}^{(a)}} e^{\beta m^a \tau_{1,u}^{(a)}}$$

$$\times \prod_{\mu>1} \left[ \sum_{\left\{\tau_u^{(a)}\right\}} e^{\frac{1}{2}\frac{\beta^2}{N_v}\sum_a \left(\sum_u \tau_u^{(a)}\right)^2 + \frac{1}{2}\frac{\beta^2}{N_v}\sum_{a\neq b} q_{ab}\left(\sum_u \tau_u^{(a)}\right)\left(\sum \tau_v^{(b)}\right)} \right]. \tag{B.3}$$

After tracing out the sum over configurations in the second line and expressing the resulting partition function as a saddle point integral, we get the following free-energy density (i.e. normalizing w.r.t. $N_v$):

$$-\beta f = i\sum_a m^a \hat{m}^a + i\sum_{a<b} q_{ab}\hat{q}_{ab} + I_S + \alpha_H \sum_a \log 2\cosh\left(\beta m^a\right) + \alpha I_E, \tag{B.4}$$

with

$$I_S = \mathbb{E}_w \log \sum_{s_a} e^{-i\sum_a \hat{m}^a s_a w - i\sum_{a<b} \hat{q}_{ab} s_i^{(a)} s_i^{(b)}}, \tag{B.5}$$

$$I_E = \log \sum_{\left\{\tau_u^{(a)}\right\}} e^{\frac{1}{2}\frac{\beta^2}{N_v}\sum_a \left(\sum_u \tau_u^{(a)}\right)^2 + \frac{1}{2}\frac{\beta^2}{N_v}\sum_{a\neq b} q_{ab}\left(\sum_u \tau_u^{(a)}\right)\left(\sum \tau_v^{(b)}\right)}. \tag{B.6}$$

### 1. Replica-symmetric (RS) ansatz

Under the following replica symmetric ansatz for the order parameters

$$m_a^\mu = m_\mu,$$
$$\hat{m}_a^\mu = i\hat{m}_\mu,$$
$$q_{ab} = q \ \text{ for } a < b,$$
$$\hat{q}_{ab} = i\hat{q} \ \text{ for } a < b,$$

we can easily rewrite the free energy (B.4). Using some integration by parts to simplify the energetic term $\mathcal{I}_E$ and the entropic one $\mathcal{I}_S$, in the limit $n \to \infty$. So the final free energy reads, in the limit $n \to 0$

$$-\beta f = -m\hat{m} + \frac{1}{2}(q-1)\hat{q} + \alpha_H \log 2\cosh\beta m + \int Dz \log 2\cosh\left(\hat{m} + \sqrt{\hat{q}}z\right)$$

$$- \frac{\alpha}{2}\log\left[1 - \alpha_H \beta^2(1-q)\right] + \alpha\alpha_H \frac{\beta^2}{2}\left[\frac{q}{1 - \beta^2(1-q)}\right], \tag{B.7}$$

which is the same expression as Eq. (16) in the main text.

## Appendix C: Derivation of free energy for Binary-Ternary RBM with adjusted biases

We consider the following Hamiltonian

$$\mathcal{H} = -\frac{1}{N_v}\sum_{\mu,i} s_i w_{i\mu} \sum_u \tau_u^{(\mu)} - \sum_{u,\mu} \theta_u \left(1 - \delta_{\tau_u^{(\mu)},0}\right),$$

where $s_i = \pm 1$ and $\tau = \{0, \pm 1\}$. In this Hamiltonian, the bias $\theta_u$ can be used to interpolate between a system where all the hidden nodes to be zero, in the limit $\theta_u \to -\infty$, or binary symmetric when $\theta_u \to \infty$. Following the same computation as for the Hopfield model, we can for instance first characterize ho the signal term (in the replicated system) can be computed

$$E_s(\boldsymbol{m}) = nN_v\alpha_H \log\left[1 + 2e^{\beta\theta}\cosh(\beta m)\right],$$

where instead of the usual log cosh, we have an additional term due to the possibility that an hidden is silenced ($\tau = 0$). In the same way, the noise term from Eq. B.2; before taking the limit $n \to 0$ but after the RS ansatz; is changed to

$$E_{\text{noise}}^{\text{RS}}(q) = \frac{P-1}{N_v} \log\left(2\int \mathcal{D}z \int \prod_a \mathcal{D}z_a \exp\sum_a \log\left[1 + 2e^{\beta\theta}\cosh\left(\frac{\beta}{\sqrt{N_v}}\left[z\sqrt{q} + z_a\sqrt{1-q}\right]\right)\right]\right). \tag{C.1}$$

Because the variables are centered, we have a cosh that we can expand up to second order as usual. We obtain

$$E_{\text{noise}}^{\text{RS}}(q) = \frac{P-1}{N_v} \log\left(2\int \mathcal{D}z \int \prod_a \mathcal{D}z_a \left(1 + 2e^{\beta\theta}\right)^{N_h} \exp\left\{\alpha_H\sigma_0(\beta\theta)\frac{\beta^2}{2N_v}\left[z\sqrt{q} + z_a\sqrt{1-q}\right]^2\right\}\right).$$

where we defined

$$\sigma_0(x) = \frac{2e^{\beta\theta}}{1 + 2e^{\beta\theta}}. \tag{C.2}$$

From this equation, we see that there is now an extra term defined by Eq. A, where the field $\theta$ can play an important role to inhibit the noise's contribution while keeping the signal non-zero.

## Appendix D: Derivation of generic free energy with arbitrary hidden prior in the high-load regime

This sections presents a detailed derivation of the equilibrium free energy of a RBM with binary visible nodes and arbitrary prior on the hidden units. Specifically, we start from the first definition (i.e. Eq. (1)) given in the main text of the Hamiltonian of such a model:

$$\mathcal{H}(\boldsymbol{s}, \boldsymbol{\tau}) = -\sum_{i\mu} s_i w_{i\mu}\tau_\mu - \sum_{i=1}^{N_v} a_i s_i - \sum_{\mu=1}^{N_h} \mathcal{U}_\mu^h(\tau_\mu), \tag{D.1}$$

where $s_i \in \{-1, 1\}$ are the visible nodes, and $\tau_\mu$ are the hidden ones. A prior on hidden units corresponds to a potential $\mathcal{U}_\mu^{(\text{h})}(\tau_\mu)$ in the above Hamiltonian. We want to compute the partition function of this model and its free energy by letting the weights to be quenched disorder with $w_{ia} \in \{-1, 1\}$. We also need to rescale the coupling term by $\sqrt{N_v}$ to be consistent in the thermodynamic limit. For simplicity, we consider no external biases in the following calculations, and the same prior over each hidden node so that $\mathcal{U}_\mu^{(\text{h})}(\tau_\mu) = \mathcal{U}^{(\text{h})}(\tau_\mu)$. Further generalizations can be made e.g. by using a either constant or random external bias, or by modeling a finite dilution in the RBM's weights, e.g. along the lines of [49]. With these settings, partition function and the quenched free energy read:

$$Z = \sum_{\boldsymbol{s}} \int d\boldsymbol{\tau} \exp\left[\frac{\beta}{\sqrt{N_v}}\sum_{i\mu} w_{i\mu}s_i\tau_\mu + \beta\sum_\mu \mathcal{U}^{(\text{h})}(\tau_\mu)\right], \tag{D.2}$$

$$f = -\frac{1}{\beta N_v}\mathbb{E}_{\boldsymbol{w}} \log Z. \tag{D.3}$$

### 1. Generic free energy computation

We start from the replicated partition function of (D.2) by splitting the signal and noise term. This time we separate the first $L$ patterns encoded in the weight matrix, corresponding to a situation where

at most $L$ nodes are strongly activated.

$$Z^n = \sum_{\boldsymbol{s}^{(a)}} \int d\boldsymbol{\tau}^{(a)} \exp\left[\frac{\beta}{\sqrt{N_{\mathrm{v}}}} \sum_{a,i} \sum_{\mu=1}^{L} w_{i\mu} s_i^{(a)} \tau_\mu^{(a)} + \frac{\beta}{\sqrt{N_{\mathrm{v}}}} \sum_{a,i,\mu>L} w_{i\mu} s_i^{(a)} \tau_\mu^{(a)} + \beta \sum_{a,\mu} \mathcal{U}^{(\mathrm{h})}\left(\tau_\mu^{(a)}\right)\right].$$
(D.4)

We now perform the average over the extensive noise:

$$\left\langle e^{\frac{\beta}{\sqrt{N_{\mathrm{v}}}} \sum_{a,i,\mu>L} w_{i\mu} s_i^{(a)} \tau_\mu^{(a)}} \right\rangle_{p(\boldsymbol{w})} = \prod_{i,\mu>L} \left\langle e^{\frac{\beta}{\sqrt{N_{\mathrm{v}}}} \sum_a w_{i\mu} s_i^{(a)} \tau_\mu^{(a)}} \right\rangle_{p(w_{i\mu})}$$

$$= \prod_{i,\mu>L} \cosh\left(\frac{\beta}{\sqrt{N_{\mathrm{v}}}} \sum_a s_i^{(a)} \tau_\mu^{(a)}\right)$$

$$\approx \exp\left[\frac{\beta^2}{2N_{\mathrm{v}}} \sum_{i,\mu>L} \left(\sum_a s_i^{(a)} \tau_\mu^{(a)}\right)^2\right]$$

$$= \exp\left[\frac{\beta^2}{2N_{\mathrm{v}}} \sum_{i,\mu>L} \sum_a \left(\tau_\mu^{(a)}\right)^2 + \frac{\beta^2}{N_{\mathrm{v}}} \sum_{i,\mu>L} \sum_{a<b} s_i^{(a)} s_i^{(b)} \tau_\mu^{(a)} \tau_\mu^{(b)}\right]. \quad \text{(D.5)}$$

Now we introduce the order parameters, namely the magnetization of the visible nodes w.r.t. the weights of the activated hidden node $\{w_{ia}\}_{i=1}^{N_{\mathrm{v}}}$ and the overlap between two visible configurations:

$$m_a^\mu = \frac{1}{N_{\mathrm{v}}} \sum_i w_{i\mu} s_i^{(a)}, \qquad \forall a \in \{1,\dots n\} \quad \text{and} \quad \forall \mu \in \{1,\dots,L\}, \quad \text{(D.6)}$$

$$q_{ab} = \frac{1}{N_{\mathrm{v}}} \sum_i s_i^{(a)} s_i^{(b)}, \qquad \forall a < b \in \{1,\dots n\}. \quad \text{(D.7)}$$

Enforcing their definitions through deltas we get

$$1 = \int \prod_{a<b} dq_{ab} d\hat{q}_{ab} \exp\left[\mathrm{i}N_{\mathrm{v}} \sum_{a<b} q_{ab} \hat{q}_{ab} - \mathrm{i} \sum_{a<b} \hat{q}_{ab} \sum_i s_i^{(a)} s_i^{(b)}\right], \quad \text{(D.8)}$$

$$1 = \int \prod_{a,\mu\leq L} dm_a^\mu d\hat{m}_a^\mu \exp\left[\mathrm{i}N_{\mathrm{v}} \sum_a m_a^\mu \hat{m}_a^\mu - \mathrm{i} \sum_a \hat{m}_a \sum_i s_i^{(a)} w_{i\mu}\right]. \quad \text{(D.9)}$$

After substituting Eqs (D.9)-(D.8) and (D.5) into (D.4) we get:

$$\langle Z^n \rangle = \int \prod_{a<b} dq_{ab} d\hat{q}_{ab} \prod_{a,\mu\leq L} dm_a^\mu d\hat{m}_a^\mu \sum_{\boldsymbol{s}^{(a)}} \int d\boldsymbol{\tau}^{(a)} \exp\left[\beta\sqrt{N_{\mathrm{v}}} \sum_{a,\mu=1}^{L} m_\mu^{(a)} \tau_\mu^{(a)} +\right.$$

$$+ \frac{1}{2}\beta^2 \sum_{a,\mu>L} \left(\tau_\mu^{(a)}\right)^2 + \beta^2 \sum_{i,\mu>L} \sum_{a<b} q_{ab} \tau_\mu^{(a)} \tau_\mu^{(b)} + \beta \sum_{a,\mu} \mathcal{U}^{(\mathrm{h})}\left(\tau_\mu^{(a)}\right) +$$

$$\left. + \mathrm{i}N_{\mathrm{v}} \sum_{a,\mu\leq L} m_a^\mu \hat{m}_a^\mu - \mathrm{i} \sum_a \hat{m}_a^\mu \sum_{i,\mu\leq L} s_i^{(a)} w_{i\mu} + \mathrm{i}N_{\mathrm{v}} \sum_{a<b} q_{ab} \hat{q}_{ab} - \mathrm{i} \sum_{a<b} \hat{q}_{ab} \sum_i s_i^{(a)} s_i^{(b)}\right]. \quad \text{(D.10)}$$

In order to have a non trivial limit for the activated hidden nodes we rescale it as $\tau_\mu^{(a)} \to \sqrt{N_{\mathrm{v}}}\tau_\mu^{(a)}$, so we set a change of variables of the type $\tau_\mu^{(a)} = \sqrt{N_{\mathrm{v}}}y_a^\mu$. After this change of variables, we rewrite (D.10) as a saddle point, defining $\alpha = N_{\mathrm{h}}/N_{\mathrm{v}}$:

$$\langle Z^n \rangle = \int \prod_{a\leq b} dq_{ab} d\hat{q}_{ab} \prod_a dm_a d\hat{m}_a \exp\left[-\beta N_{\mathrm{v}} f\right], \quad \text{(D.11)}$$

$$f = -\frac{\mathrm{i}}{\beta} \sum_{a,\mu\leq L} m_a^\mu \hat{m}_a^\mu - \frac{\mathrm{i}}{\beta} \sum_{a\leq b} q_{ab} \hat{q}_{ab} - \frac{1}{\beta}\mathcal{I}_S - \frac{\alpha}{\beta}\mathcal{I}_E - \frac{1}{\beta}\mathcal{I}_R. \quad \text{(D.12)}$$

In the above expression we defined for convenience three quantities, an entropic term ($\mathcal{I}_S$) depending on the values of visible nodes, and two energetic terms, one associated to the signal term $\mathcal{I}_R$ and another one due to all the other hidden nodes which are not activated $\mathcal{I}_E$. All of them are defined below:

$$\mathcal{I}_S = \mathbb{E}_{\boldsymbol{w}} \log \sum_{\boldsymbol{s}} \exp\left[ -\mathrm{i} \sum_{a,\mu\leq L} \hat{m}_a^\mu w_\mu s_a - \mathrm{i} \sum_{a\leq b} \hat{q}_{ab} s_a s_b \right], \tag{D.13}$$

$$\mathcal{I}_R = \frac{1}{N_{\mathrm{v}}} \log \int \prod_{a,\mu\leq L} dy_a^\mu \exp\left[ \beta N_{\mathrm{v}} \sum_{a,\mu\leq L} m_a^\mu y_a^\mu + \beta \sum_{a,\mu\leq L} \mathcal{U}^{(\mathrm{h})}\left(\sqrt{N_{\mathrm{v}}} y_a^\mu\right) \right], \tag{D.14}$$

$$\mathcal{I}_E = \log \int \prod_a d\tau_a \exp\left[ \frac{\beta^2}{2} \sum_a \tau_a^2 + \beta^2 \sum_{a<b} q_{ab} \tau_a \tau_b + \beta \sum_a \mathcal{U}^{(\mathrm{h})}\left(\tau_a\right) \right]. \tag{D.15}$$

Note that the entropic term just depends on the visible nodes, so not on the potential $\mathcal{U}^{(\mathrm{h})}$. Notice that the signal term is factorized over both $a$ and $\mu \leq L$

$$\mathcal{I}_R = \frac{1}{N_{\mathrm{v}}} \sum_{a,\mu\leq L} \log \int dy \exp\left[ \beta N_{\mathrm{v}} y m_a^\mu + \beta \mathcal{U}^{(\mathrm{h})}\left(\sqrt{N_{\mathrm{v}}} y\right) \right]. \tag{D.16}$$

## 2. Replica symmetry

Assume now RS in the following form

$$
\begin{aligned}
m_a^\mu &= m_\mu, \\
\hat{m}_a^\mu &= \mathrm{i}\beta \hat{m}_\mu, \\
q_{ab} &= q\left(1 - \delta_{ab}\right) \quad \text{for} \quad a < b, \\
\hat{q}_{ab} &= \mathrm{i}\alpha\beta^2 \hat{q} \quad \text{for} \quad a < b,
\end{aligned}
$$

and rewrite the three integrals (D.13)(D.14)(D.15). Derivation steps are given below.

*a. Entropic term $\mathcal{I}_S$*

$$
\begin{aligned}
\mathcal{I}_S &= \mathbb{E}_{\boldsymbol{w}} \log \sum_{\boldsymbol{s}} \exp\left[ \beta\left(\hat{\boldsymbol{m}}\cdot\boldsymbol{w}\right) \sum_a s_a + \alpha\beta^2 \hat{q} \sum_{a<b} s_a s_b \right] \\
&= \mathbb{E}_{\boldsymbol{w}} \log \sum_{\boldsymbol{s}} \exp\left\{ \beta\left(\hat{\boldsymbol{m}}\cdot\boldsymbol{w}\right) \sum_a s_a + \frac{1}{2}\alpha\beta^2 \hat{q} \left[ \left(\sum_a s_a\right)^2 - n \right] \right\} \\
&= -\frac{n}{2}\alpha\beta^2 \hat{q} + \mathbb{E}_{\boldsymbol{w}} \log \sum_{\boldsymbol{s}} \int Dz \exp\left\{ \beta\left(\hat{\boldsymbol{m}}\cdot\boldsymbol{w}\right) \sum_a s_a + \beta\sqrt{\alpha\hat{q}} z \sum_a s_a \right\} \\
&= -\frac{n}{2}\alpha\beta^2 \hat{q} + \mathbb{E}_{\boldsymbol{w}} \log \int Dz\, 2^n \cosh^n\left[ \beta\left(\hat{\boldsymbol{m}}\cdot\boldsymbol{w}\right) + \beta\sqrt{\alpha\hat{q}} z \right] \\
&\approx -\frac{n}{2}\alpha\beta^2 \hat{q} + n\mathbb{E}_{\boldsymbol{w}} \int Dz \log 2\cosh\left[ \beta\left(\hat{\boldsymbol{m}}\cdot\boldsymbol{w}\right) + \beta\sqrt{\alpha\hat{q}} z \right]. \tag{D.17}
\end{aligned}
$$

*b. Energetic, noise term*

$$\mathcal{I}_E = \log \int \prod_a d\tau_a \exp\left[\frac{\beta^2}{2}\sum_a \tau_a^2 + \beta^2 q \sum_{a<b} \tau_a \tau_b + \beta \sum_a \mathcal{U}^{(\mathrm{h})}(\tau_a)\right]$$

$$= \log \int \prod_a d\tau_a \exp\left\{\frac{\beta^2}{2}\sum_a \tau_a^2 + \frac{1}{2}\beta^2 q\left[\left(\sum_a \tau_a\right)^2 - \sum_a \tau_a^2\right] + \beta \sum_a \mathcal{U}^{(\mathrm{h})}(\tau_a)\right\}$$

$$= \log \int \prod_a d\tau_a \int Dz \exp\left[\frac{\beta^2(1-q)}{2}\sum_a \tau_a^2 + \beta\sqrt{q}z\sum_a \tau_a + \beta \sum_a \mathcal{U}^{(\mathrm{h})}(\tau_a)\right]$$

$$= \log \int Dz \left\{\int d\tau \exp\left[\frac{\beta^2(1-q)}{2}\tau^2 + \beta\sqrt{q}z\tau + \beta\mathcal{U}^{(\mathrm{h})}(\tau)\right]\right\}^n$$

$$= n \int Dz \log \int d\tau \exp\left[\frac{\beta^2(1-q)}{2}\tau^2 + \beta\sqrt{q}z\tau + \beta\mathcal{U}^{(\mathrm{h})}(\tau)\right]. \tag{D.18}$$

*c. Energetic, signal term*

$$\mathcal{I}_R = \frac{n}{N_{\mathrm{v}}}\sum_{\mu\leq L}\log\int dy\exp\left[\beta N_{\mathrm{v}}y m_\mu + \beta\mathcal{U}^{(\mathrm{h})}\left(\sqrt{N_{\mathrm{v}}}y\right)\right] \tag{D.19}$$

After the above simplifications, we can write the the free energy in the limit $n\to 0$ as

$$f = +\sum_{\mu\leq L}m_\mu \hat{m}_\mu + \frac{\alpha\beta}{2}\hat{q}(1-q) - \frac{1}{\beta N_{\mathrm{v}}}\sum_{\mu\leq L}\log\int dy\exp\left[\beta N_{\mathrm{v}}y m_\mu + \beta\mathcal{U}^{(\mathrm{h})}\left(\sqrt{N_{\mathrm{v}}}y\right)\right]$$

$$- \frac{1}{\beta}\mathbb{E}_{\boldsymbol{w}}\int Dz\log 2\cosh\left[\beta(\hat{\boldsymbol{m}}\cdot\boldsymbol{w}) + \beta\sqrt{\alpha\hat{q}}z\right]$$

$$- \frac{\alpha}{\beta}\int Dz\log\int d\tau\exp\left[\frac{\beta^2(1-q)}{2}\tau^2 + \beta\sqrt{q}z\tau + \beta\mathcal{U}^{(\mathrm{h})}(\tau)\right]. \tag{D.20}$$

This is a generic expression holding for any potential on the hidden nodes. In particular, the hidden potential $\mathcal{U}^{(\mathrm{h})}$ affects the signal $\mathcal{I}_R$ and noise $\mathcal{I}_E$ energetic terms.

## Appendix E: Specifying Potential RELU

For the moment we consider a ReLU potential with an external field $\eta$. The expression of the potential is [49]

$$\mathcal{U}(\tau) = \begin{cases} -\frac{\lambda}{2}\tau^2 + \eta\tau & \tau > 0, \\ \infty & \text{otherwise.} \end{cases} \tag{E.1}$$

which corresponds to a *truncated* Gaussian prior in the interval $(0, +\infty)$. We specify below the expressions of the energetic terms with the potential (E.1)

*a. Signal term $\mathcal{I}_R$*

$$\mathcal{I}_R = \frac{1}{N_{\mathrm{v}}}\sum_{\mu\leq L}\log\int dy\exp\left[\beta N_{\mathrm{v}}y m_\mu + \beta\mathcal{U}\left(\sqrt{N_{\mathrm{v}}}y\right)\right]$$

$$= \frac{1}{N_{\mathrm{v}}}\sum_{\mu\leq L}\log\int_0^\infty dy\exp\left[\beta N_{\mathrm{v}}m_\mu y - \beta\lambda N_{\mathrm{v}}\frac{y^2}{2} + O\left(\sqrt{N_{\mathrm{v}}}\right)\right]$$

$$= \frac{1}{N_{\mathrm{v}}}\sum_{\mu\leq L}\log\int_0^\infty dy\exp\left[\beta N_{\mathrm{v}}m_\mu y - \beta\lambda N_{\mathrm{v}}\frac{y^2}{2}\right]$$

$$= \frac{1}{N_{\mathrm{v}}}\sum_{\mu\leq L}\log\left\{\exp\left[\frac{\beta N_{\mathrm{v}}m_\mu^2}{2\lambda}\right]\sqrt{\frac{\pi}{2\beta\lambda N_{\mathrm{v}}}}\left[1 + \mathrm{erf}\frac{\beta N_{\mathrm{v}}m_\mu}{\sqrt{2\beta\lambda N_{\mathrm{v}}}}\right]\right\}$$

$$= \sum_{\mu\leq L}\left\{\frac{\beta}{2\lambda}m_\mu^2 - \frac{1}{2N_{\mathrm{v}}}\log\frac{2\beta\lambda N_{\mathrm{v}}}{\pi} + \frac{1}{N_{\mathrm{v}}}\log\left[1 + \mathrm{erf}\frac{\beta N_{\mathrm{v}}m_\mu}{\sqrt{2\beta\lambda N_{\mathrm{v}}}}\right]\right\}.$$

In the limit $N_\mathrm{v} \to \infty$, the second term cancels and the third one will depend on the sign of $m$: specifically, it will be 0 when $m_\mu > 0$. When $m_\mu < 0$ there is a non vanishing limit computed by using the asymptotic expansion of the complementary error function, and we get the following expression:

$$\mathcal{I}_R = \frac{\beta}{2\lambda} \sum_{\mu \leq L} m_\mu^2 \Theta\left(m_\mu\right), \tag{E.2}$$

where $\Theta\left(m\right)$ is the Heaviside step function.

    b.   *Energetic term $\mathcal{I}_E$*

$$\begin{aligned}
\mathcal{I}_E &= \int Dz \log \int d\tau \exp\left[\frac{\beta^2}{2}\left(1-q\right)\tau^2 + \beta\sqrt{q}z\tau + \beta\mathcal{U}^{(\mathrm{h})}\left(\tau\right)\right] \\
&= \int Dz \log \int_0^\infty d\tau \exp\left[\frac{\beta^2}{2}\left(1-q\right)\tau^2 + \beta\sqrt{q}z\tau - \frac{\beta\lambda}{2}\tau^2 + \beta\eta\tau\right] \\
&\int Dz \log \int_0^\infty d\tau \exp\left[\frac{\beta^2\left(1-q\right)}{2}\tau^2 + \beta\sqrt{q}z\tau - \frac{\beta\lambda}{2}\tau^2 + \beta\eta\tau\right] \\
&= \int Dz \log \int_0^\infty d\tau \exp\left[-\frac{\beta\Delta}{2}\tau^2 + \left(\beta\sqrt{q}z + \beta\eta\right)\tau\right] \\
&= \int Dz \log \left\{\sqrt{\frac{2\pi}{\beta\Delta}} \exp\left[\frac{1}{2\beta\Delta}\left(\beta\sqrt{q}z + \beta\eta\right)^2\right] H\left[-\frac{\left(\beta\sqrt{q}z + \beta\eta\right)}{\sqrt{\beta\Delta}}\right]\right\} \\
&= -\frac{1}{2}\log\Delta + \int Dz \left[\frac{\beta}{2\Delta}\left(\sqrt{q}z + \eta\right)^2\right] + \int Dz \log H\left[-\frac{\beta\left(\sqrt{q}z + \eta\right)}{\sqrt{\beta\Delta}}\right] \\
&= -\frac{1}{2}\log\Delta + \frac{\beta\left(q+\eta^2\right)}{2\Delta} + \int Dz \log H\left[-\sqrt{\frac{\beta}{\Delta}}\left(\sqrt{q}z + \eta\right)\right] \\
&= -\frac{1}{2}\log\Delta + \frac{\beta}{2\Delta}\left(q+\eta^2\right) + \int Dz \log H\left[\sqrt{\frac{\beta}{\Delta}}\left(\sqrt{q}z - \eta\right)\right], \tag{E.3}
\end{aligned}$$

where we defined $\Delta = \lambda - \beta\left(1-q\right)$ and the function $H\left(x\right) = \frac{1}{2}\mathrm{erfc}\left(\frac{x}{\sqrt{2}}\right)$. In the last line we exploited the fact that $\int Dz f\left(-z\right) = \int Dz f\left(z\right)$. Using these two results, the free energy in general support of the RBM with ReLU hidden layer is given by

$$\begin{aligned}
f &= \sum_{\mu \leq L} m_\mu \hat{m}_\mu - \frac{1}{2\lambda} \sum_{\mu \leq L} m_\mu^2 \Theta\left(m_\mu\right) + \frac{\alpha\beta}{2}\hat{q}\left(1-q\right) \\
&+ \frac{\alpha}{2\beta}\log\Delta - \frac{\alpha}{2\Delta}\left(q+\eta^2\right) - \frac{\alpha}{\beta}\int Dz \log H\left(az + b\right) \\
&\frac{1}{\beta}\mathbb{E}_{\boldsymbol{w}} \int Dz \log 2\cosh\left[\beta\left(\hat{\boldsymbol{m}}\cdot\boldsymbol{w}\right) + \beta\sqrt{\alpha\hat{q}}z\right], \tag{E.4}
\end{aligned}$$

with $b = -\eta\sqrt{\frac{\beta}{\Delta}}$ and $a = \sqrt{\frac{\beta q}{\Delta}}$; $\Delta = \lambda - \beta\left(1-q\right)$.