# Peer review of "On the role of non-linear latent features in bipartite generative neural networks"

_SciPost Physics_

## Round 1 · Referee Report · Anonymous (Referee 1) · 2025-7-1

Report

In their paper entitled “On the role of non-linear latent features in bipartite generative neural networks” the authors perform a thoroughly analysis or restricted Boltzmann Machines (RBM) exploring how their memory retrieval capabilities vary when changing the prior and depending on the number of latent features. This investigation encompasses both analytical and numerical approaches, finding overall a good agreement. The presentation is particularly clear and include a nice overview on the subject. Overall I find that this work provides a valuable contribution on a paradigmatic model of neural networks and deserves publication in SciPost. I have only minor issues that I would ask the authors to consider before recommending publication.

  • In the abstract, the authors state “bipartite energy-based neural networks, namely Restricted Boltzmann Machines”, maybe I am wrong, but that class of networks contains several models beyond RBS, for instance bidirectional associative memories. If the authors agree, maybe they could reshuffle the sentence.

  • As far as I could see, calculations are carried out under the RS ansatz, which is pretty standard in the literature for this kind of models. Yet, given the rather general introduction, potentially attracting a wide audience, and consistently with the journal mission, I believe that a short discussion on the possible effects of RSB, either qualitative or quantitative, would further strengthen the presentation (see e.g., Agliari et al. “A transport equation approach for deep neural networks with quenched random weights”, and Fachechi et al. “ Fundamental operating regimes, hyper-parameter fine-tuning and glassiness: towards an interpretable replica-theory for trained restricted Boltzmann machines ” for different approaches to RSB phenomena). This would possibly allow the authors to comment the dashed line appearing in the phase diagram of Fig. 1.

  • The introduction is generally well written, but the motivation for comparing binary and ReLU hidden units could be stated more clearly at the outset for readers less familiar with RBMs. Analogously, terms like “recall phase,” “ferromagnetic solution,” “spinodal point” and “signal” are occasionally used without immediate definition. It would help accessibility to briefly recall these for interdisciplinary readers.

  • The MNIST training experiment is illustrative, but a slightly more detailed discussion (perhaps a few lines) on how the redundancy in learned features relates to the theory would be welcome.

To conclude, I find that this is a high-quality, theoretically rich, and timely contribution that will interest both statistical physicists and machine learning researchers. In my opinion, it fits well within SciPost Physics' mission and I recommend its publication, subject to the minor issued detailed above.

Recommendation

Ask for minor revision

  • validity: -
  • significance: -
  • originality: -
  • clarity: -
  • formatting: -
  • grammar: -

Author:  Aurélien Decelle  on 2025-09-16  [id 5827]

(in reply to Report 1 on 2025-07-01)
Category:
answer to question

We thank the referee for his careful reading of the manuscript, his positive feedback and comments. We answer to all the comments below.

  • We believe that bipartite energy-based NN is indeed somehow a synonym for the Restricted Boltzmann Machine, but to avoid possible cases that could be described differently, we rephrase the first sentence as "We investigate the phase diagram and memory retrieval capabilities of Restricted Boltzmann Machines (RBMs), an archetypal model of bipartite energy-based neural networks, as a function of the prior distribution imposed on their hidden units—including binary, multi-state, and ReLU-like activations.". Also, bidirectional associative memories can indeed be expressed as an RBM where the weights have some particular structure, but I guess this is debatable.

  • The referee is right: all computations are made within the RS assumption. Following the referee's suggestion, we add a small paragraph on 1RSB in the Hopfield model in page 5. While, as mentioned in the paragraph, our work does not particularly focus on this phase, this is not referred to in the article. However, the conclusion now discusses the 1RSB relation to the learning phase diagram, as computed in Fachechi et al., 2025.

  • A new paragraph was added at the end of the introduction to give more context and motivations on why studying ReLU hidden units in RBMs in page 3. As suggested, we also now take care to define the various terms related to the different phases when presenting it in page 4 in Sect. 2.

  • We expanded our discussion of the MNIST experiment at the end of Sect. 3 to clarify the learning behavior of the RBM.

---

## Round 1 · Referee Report · Anonymous (Referee 2) · 2025-8-3

Strengths

Understanding how generative models learn from data and how they are able to generate new data is a very interesting and timely question. In recent years, the machine learning community has seen impressive technical progress, which unfortunately is not supported yet by a solid theoretical understanding.

RBM are a relatively simple unsupervised model. By focusing on RBM, this work is a contribution towards reaching a better understanding of generative models in general. In particular, RBM have proven amenable to analysis by classic statistical physics approaches, as this paper (and others before) have expertly demonstrated.

This paper presents a statistical physics analysis of the RBM with different non-linear hidden unit potentials, while the visible units remain binary. They study the impact of the hidden unit potential choice on the capacity of the model to recall random stored patterns in the weights.

The paper is well-written and pedagogical.

Weaknesses

The authors study random patterns. Real data is likely to exhibit complex structure, which is then reflected in the weights of a trained RBM. This assumption therefore limits the impact of the results of this work. However, similar assumptions are common in statistical physics approaches to neural networks. Moreover, I realize that overcoming this assumption may entail substantial challenges.

Report

I believe this paper is valuable addition to the literature. It is well written and easy to follow. I have only a few suggestions, regarding typos or some paragraphs that could be slightly rewritten to enhance clarity. Besides those points, I will recommend publication.

I would also like to point out a couple of extensions that I would find interesting, but which the authors can also choose to leave for future work:

  • This work has focused on visible binary units, while treating various hidden unit potentials. I think one interesting missing case the authors could potentially treat is an RBM where both visible and hidden units are continuous. Beyond the the Gaussian-Gaussian case analyzed by Karakida et al 2016, it would be interesting to look at other combinations, like ReLU-Gaussian, or ReLU-ReLU. This could be relevant for some kinds of data, such as images.

  • The authors have focused on random patterns. How does the situation change for correlated patterns? Does recall become possible, even for binary hidden units without bias?

Requested changes

  • Reference [22] and [24] are duplicates.

  • From the remark just below Eq. (5), I presume the authors intended to eliminate w_i from the second line of Eq. (5), but forgot to do so.

  • The authors claim no recall phase is possible with binary hidden units (without bias), unless they are replicated extensively. However the simple case with a single hidden unit presented in Eq. (4) is evidently able to recall the single pattern, at all temperatures. Did the authors intend to say that the absence of a recall phase applies only to extensive patterns? Perhaps I'm misunderstanding something here, but I think the authors could rephrase a bit better this message, to make it clearer and resolve the (apparent) contradiction.

  • It seems that in the last passage of Eq. (B.2), the authors replace logcosh(x) with x^2/2, dropping higher-order terms in the Taylor expansion of logcosh. However, since the sum over 'u' goes over replicated hidden units, attached to the same pattern, they are likely to be correlated, which implies that this sum over 'u' is not necessarily small. In this case, the quadratic approximation of the logcosh is not warranted. Can the authors clarify this point? How do they justify the smallness of the logcosh argument here and the ensuing quadratic approximation ? Is this exact in some sense, or the analysis is only approximate? I believe the authors should explain better this step. The same applies to (D.5).

  • Regarding the "final remark on the ReLU potential" (just before Conclusion), the authors' suggestion of a hidden unit with a finite mass at 0 sounds similar to the Spike and Slab RBM (Courville et al 2011). If related, the authors could cite this work and perhaps add a sentence about it at this point.

  • Typo on page 19, "first characterize ho the signal term" .

  • The authors should comment on the validity of the replica-symmetry (RS) assumption.

  • Please provide the code for the simulations done in the paper.

  • This paper focus on equilibrium statistical mechanics of the RBM. But the phase diagram surely has interesting dynamical consequences that impact training or sampling. Could the authors comment/speculate on this ?

Recommendation

Publish (meets expectations and criteria for this Journal)

  • validity: high
  • significance: good
  • originality: good
  • clarity: high
  • formatting: excellent
  • grammar: excellent

Author:  Aurélien Decelle  on 2025-09-16  [id 5828]

(in reply to Report 2 on 2025-08-03)
Category:
answer to question

We thank the referee for his careful reading of the manuscript, his positive feedbacks and comments. We carefully answer to all comments and remarks below.

The referee mentioned as weaknesses:

The authors study random patterns. Real data is likely to exhibit complex structure, which is then reflected in the weights of a trained RBM. This assumption therefore limits the impact of the results of this work. However, similar assumptions are common in statistical physics approaches to neural networks. Moreover, I realize that overcoming this assumption may entail substantial challenges.

Our answer: We agree with the referee that our theoretical study mainly focus on random patterns, letting aside real data. However, we believe that our example on the MNIST dataset provides a clear illustration of how the theoretical setting connects with a concrete learning task, and the results are in good agreement. In addition, an analytical treatment considering real data would be extremely challenging.

On the report, the following comment was made:

  • This work has focused on visible binary units, while treating various hidden unit potentials. I think one interesting missing case the authors could potentially treat is an RBM where both visible and hidden units are continuous. Beyond the the Gaussian-Gaussian case analyzed by Karakida et al 2016, it would be interesting to look at other combinations, like ReLU-Gaussian, or ReLU-ReLU. This could be relevant for some kinds of data, such as images.
  • The authors have focused on random patterns. How does the situation change for correlated patterns? Does recall become possible, even for binary hidden units without bias?

Our answers:

  • The referee's suggestion is indeed interesting. Our first comment is that, in the work of Courville et al. mentioned in the requested change, the authors claim that, when it comes to modelling images, the "spike and slab" RBM with continuous variables is a suitable model. Since their article shows the efficiency of such an approach, we can imagine that there could be a recall phase in this model, as was demonstrated for the Continuous Hopfield model with an additional quatric potential as in Bollé et al (J. of Phys A 2003). A full replica analysis of this setting would however come with substantial additional work, which we prefer to leave for future work in order to keep the present article shorter and coherent. Yet we add the reference in the main text for completeness.
  • We agree with the referee that the case of correlated patterns would be of interest, particularly using for instance the hidden manifold construction (Goldt et al., 2020; Negri et al., 2023). We however believe it would dilute our message about the non-linearity of the hidden activations. Other correlated constructions are possible, however most of them concern the low-load regime. We thus believe that this part should be left for future studies.

We agree with the referee that these are important avenues for future research, and we now discuss them in the Conclusions and Perspectives section.

Concerning the requested changes:

  • We merged the references 22 and 24.

  • We corrected Eq. 5 by removing the extra $w_i$.

  • We clarified more precisely what we meant: the polarization at all temperature is very bad for a machine learning model. We consequently added the following sentence in page 6 of the main text: "the RBM with hidden binary units provides a different phenomenology than the classical paramagnetic-ferromagnetic phase transition in temperature. In this case, the system remains polarized at all temperatures. This is clearly not a good behavior for a Machine Learning model as the system will quickly polarize towards a pattern that is uncorrelated with the dataset and break the learning."

  • In the Eqs. B.2, the argument of the log cosh is expanded assuming that $\sum_u \tau_{\mu,u}^{(a)}$ is not extensive. We are following here a classical derivation of the Hopfield model (under the replica approach) where only one pattern is recalled. The idea is that the spin configuration should have some extensive overlap with one randomly choosen pattern (here the first one), the activation of the other hidden nodes cannot be too strong and therefore will be of order $\sim \mathcal{O}(\sqrt{\alpha_H N_v})$. This is confirmed by the fact that the thermodynamics limit is well-defined. The same applies for Eqs. D.5. We add the following sentence right after the concerned equations: "where the argument of the $\log \cosh$ can be expanded assuming that we are recalling one pattern and therefore the activation of the hidden nodes for the other patterns is of order of $\sim \mathcal{O}(\sqrt{\alpha_H N_v})$".

  • We thank the referee for mentioning this reference that we missed. We added a sentence in the conclusion indicating the similarity between the two models: the one of Courville et al. is using a spike variable and a Gaussian distribution.

  • We corrected the typo on page 19.

  • This point is treated in the answer to Referee 1.

  • The code to train the RBM for the MNIST case (Fig. 2) can be found at https://github.com/DsysDML/rbms. The code to compute the phase diagram (Figs. 3,5) is using a very basic Monte Carlo method from the alternate sampling scheme, usual for RBMs. Finally, the phase diagram is obtained by solving the self-consistent replica equations numerically, using the Julia library: https://github.com/giovact/FixedPointSolver .

  • The referee is correct; the static phase diagram has a strong impact on learning. We added a discussion about this in the conclusion. The phase diagram that we derived can be related to a case of Bayesian learning. We now also comment on the fact that the learning phase diagram, as derived in Fachechi et al. 2025 and Cossio et al. 2024, should be impacted by the prior on the hidden nodes, and that it would be interesting, yet left for future work, to understand how it is impacted by the prior.

---

## Editorial Decision

resubmitted